Clinical Science and Epidemiology

# End-to-End Protocol for the Detection of SARS-CoV-2 from Built Environments

Ceth W. Parker,[a] Nitin Singh,[a] Scott Tighe,[b] Adriana Blachowicz,[a] Jason M. Wood,[a] Arman Seuylemezian,[a] Parag Vaishampayan,[a] Camilla Urbaniak,[a,c] Ryan Hendrickson,[a] Pheobe Laaguiby,[b] Kevin Clark,[a] Brian G. Clement,[a] Niamh B. O'Hara,[d,e] Mara Couto-Rodriguez,[d] Daniela Bezdan,[f,g] Christopher E. Mason,[f] Kasthuri Venkateswaran[a]

[a]NASA Jet Propulsion Laboratory, California Institute of Technology, Pasadena, California, USA
[b]Vermont Integrative Genomics Resource, Larner College of Medicine, The University of Vermont, Burlington, Vermont, USA
[c]ZIN Technologies Inc., Middleburg Heights, Ohio, USA
[d]Biotia, New York, New York, USA
[e]SUNY Downstate Health Sciences University, Brooklyn, New York, USA
[f]Weill Medical College of Cornell University, New York, New York, USA
[g]Institute of Medical Virology and Epidemiology of Viral Diseases, University Hospital, Tubingen, Germany

Ceth W. Parker, Nitin Singh, and Scott Tighe contributed equally. The placement is based on alphabetical order.

**ABSTRACT** Severe acute respiratory syndrome coronavirus 2 (SARS-CoV-2), the virus that causes coronavirus disease 2019, is a respiratory virus primarily transmitted person to person through inhalation of droplets or aerosols, laden with viral particles. However, as recent studies have shown, virions can remain infectious for up to 72 h on surfaces, which can lead to transmission through contact. Thus, a comprehensive study was conducted to determine the efficiency of protocols to recover SARS-CoV-2 from surfaces in built environments. This end-to-end (E2E) study showed that the effective combination for monitoring SARS-CoV-2 on surfaces includes using an Isohelix swab collection tool, DNA/RNA Shield as a preservative, an automated system for RNA extraction, and reverse transcriptase quantitative PCR (RT-qPCR) as the detection assay. Using this E2E approach, this study showed that, in some cases, noninfectious viral fragments of SARS-CoV-2 persisted on surfaces for as long as 8 days even after bleach treatment. Additionally, debris associated with specific built environment surfaces appeared to inhibit and negatively impact the recovery of RNA; Amerstat demonstrated the highest inhibition (>90%) when challenged with an inactivated viral control. Overall, it was determined that this E2E protocol required a minimum of 1,000 viral particles per 25 cm$^2$ to successfully detect virus from test surfaces. Despite our findings of viral fragment longevity on surfaces, when this method was employed to evaluate 368 samples collected from various built environmental surfaces, all samples tested negative, indicating that the surfaces were either void of virus or below the detection limit of the assay.

**IMPORTANCE** The ongoing severe acute respiratory syndrome coronavirus 2 (SARS-CoV-2) (the virus responsible for coronavirus disease 2019 [COVID-19]) pandemic has led to a global slowdown with far-reaching financial and social impacts. The SARS-CoV-2 respiratory virus is primarily transmitted from person to person through inhalation of infected droplets or aerosols. However, some studies have shown that virions can remain infectious on surfaces for days and can lead to human infection from contact with infected surfaces. Thus, a comprehensive study was conducted to determine the efficiency of protocols to recover SARS-CoV-2 from surfaces in built environments. This end-to-end study showed that the effective combination for monitoring SARS-CoV-2 on surfaces required a minimum of 1,000 viral particles per 25 cm$^2$ to successfully detect virus from surfaces. This comprehensive study can pro-

**Ad Hoc Peer Reviewer** Fuqing Wu, MIT

Address correspondence to Kasthuri Venkateswaran, kasthuri.j.venkateswaran@jpl.nasa.gov.

This new end-to-end study showed that an effective combination of monitoring SARS-CoV-2 on built environment surfaces required a minimum of 1,000 virions per 25 cm2 to successfully detect the virus.#COVID-19 #SARS-Cov-2 #E2E #End-to-End #Fomites

vide valuable information regarding surface monitoring of various materials as well as the capacity to retain viral RNA and allow for effective disinfection.

**KEYWORDS** COVID-19, SARS-CoV-2, surface sampling, built environments, end-to-end, fomites, coronavirus, high-touch surface, LAMP, RT-qPCR

The ongoing coronavirus disease 2019 (COVID-19) pandemic is caused by severe acute respiratory syndrome coronavirus 2 (SARS-CoV-2) (1), which was first identified in Wuhan, China, in December 2019. The World Health Organization (WHO) declared it a Public Health Emergency of international concern on 30 January 2020, and then a pandemic on 11 March 2020. The high infection rate and rapid spread have caused global, social, and economic disruption (2), including postponement of many sporting, religious, political, and cultural events, as well as the closure of nonessential businesses, schools, and universities worldwide across 160 countries (3).

A primary goal set forth by the government of the United States has been to keep essential businesses (grocery stores, hospitals, gas stations, etc.) open while protecting staff and patrons with as little disruption as possible given the severity of the situation. The current model suggests that the main route of infection is person to person through inhalation of aerosolized droplets containing the virus (4). In addition, the use of masks, maintaining physical distancing, avoiding touching one's face, and washing hands have all been identified as important factors in preventing transmission (5). However, since SARS-CoV-2 can remain infective for hours to days on surfaces (6), it is possible to transmit and contract the virus by coming in contact with contaminated surfaces (7). When infected individuals inadvertently carry SARS-CoV-2 into built environments, the infection may spread between individuals via fomites, compromising the ability of workers to continue normal operations and activities. Therefore, disinfection and cleaning regimens have been established by most organizations as a precautionary measure to safeguard against viral transmission (8).

Since SARS-CoV-2 is fatal and a worldwide concern, the Centers for Disease Control and Prevention (CDC) and National Institutes of Health (NIH) have issued directives that molecularly (RNA) based detection be applied to clinical specimens (4, 5), but no such policies have been set for environmental monitoring (6). Since the risk of infection from contaminated surfaces is of serious concern, the need for environmental surface monitoring, along with understanding the effectiveness of cleaning and disinfection, is critical. In this study, we outline a comprehensive approach to characterize and develop an effective environmental monitoring methodology that can be used to better understand viral persistence in built environments and aid in the elimination of virus.

The ability to collect and analyze samples is fundamental to any microbial monitoring analysis. During this study, a noninfectious and replication-deficient virus was used as a surrogate for the SARS-CoV-2 virus to inoculate representative test surfaces and analyzed for recovery efficiency. Several sampling strategies were evaluated for collecting samples from various materials. Experimental parameters such as the method of viral inoculation of each surface type, collection and transport, and analysis techniques were used to determine viral recovery efficiency, total biomass, species-specific recovery, background contaminant levels, inhibitory factors, and sampling and detection anomalies.

The overall objective of the study was to develop a standardized end-to-end (E2E) protocol for the detection of SARS-CoV-2 from built environmental surfaces and to determine the minimum number of RNA copies needed on fomites to positively detect virus within the limit of detection of our assay. This study included collecting ~400 samples from seven surface types common to materials found in the built environment and measuring the recovery efficiency of the surrogate SARS-CoV-2 virus. After establishing the E2E protocol, further reproducibility studies were conducted by a second laboratory for verification.

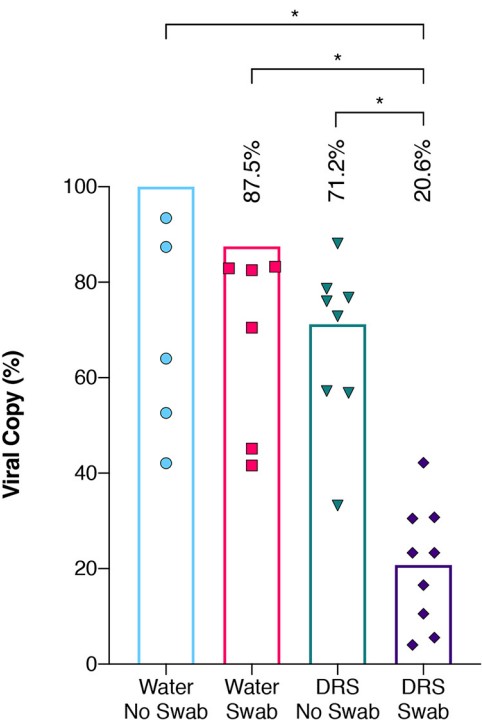

**FIG 1** Influence of swab and DRS on viral RNA extraction efficiency. Equal quantities of the inactivated AccuPlex viral particles were extracted using a variety of initial extraction conditions and then quantified using RT-qPCR assay. The extraction conditions encompassed water with no swab (blue circles), water with Isohelix swab (red squares), DNA/RNA Shield (DRS) with no swab (green inverted triangles), and DRS with Isohelix swab (purple diamonds). Each extraction condition was then divided by the average copy numbers generated from the water with no swab (theoretical highest yield) to get percent recovery and plotted with columns representing their mean percentage. Welch's $t$ test was used to determine significant differences between extraction conditions, and significance ($P < 0.05$) is denoted by asterisks.

## RESULTS

**Efficiency and influence of swab and DRS solution on viral extraction.** To determine the impact of the swabs and DNA/RNA Shield (DRS) transfer medium on the percent recovery of viral particles, the SeraCare AccuPlex SARS-CoV-2 reference material was added to tubes containing water and DRS solution, either directly into the tubes or inoculated onto swabs first, before RNA extraction followed by reverse transcriptase quantitative PCR (RT-qPCR). The resulting viral copy numbers were then compared and computed to understand the effects of swabs and DRS solution individually, along with the combined impact of swabs and DRS, in the recovery of viral particles (Fig. 1). The RNA copies detected from AccuPlex placed into the water suspension (no swab) were used as the 100% positive copy number reference to calculate other combinations. For practical applications, swabs should be placed either in water or in a transport medium like DRS so that samples can be transported and processed in the laboratory. Relative to AccuPlex in water (no swab), there was a 12% loss of viral load when AccuPlex solution was soaked on the swab before being placed in water (swab effect). Similarly, when AccuPlex was placed directly into DRS instead of water (no swab), the recovery was 71% (DRS effect). The double effect of the swab and DRS on viral recovery was significantly less ($P = 0.0008$), with only 21% recovery (Fig. 1).

**RNA extraction efficiency.** An automated RNA extraction system was compared to a manual extraction where AccuPlex was inoculated on swabs containing DRS (Fig. 2). There were no significant differences between the automated system and the manual extraction method. Subsequently, several viral transport media were also compared with water. No significant differences were observed between the three different viral transport media (Fig. 2). All tested combinations yielded between 183 and 204 nucleocapsid N1 fragment copies per 5 $\mu$l RNA extract. To characterize the extraction

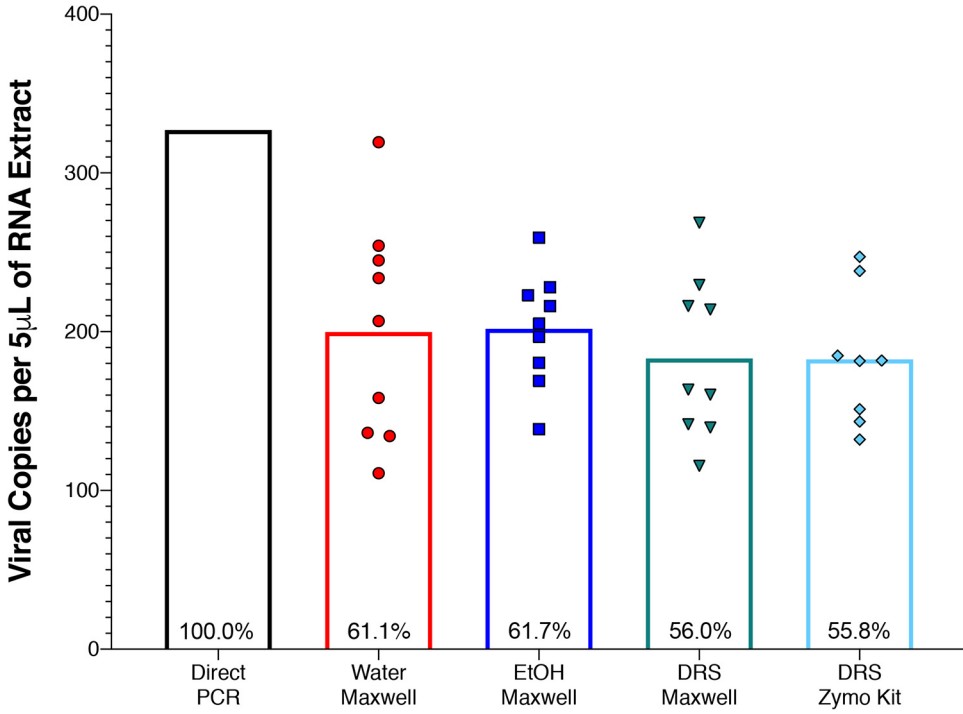

**FIG 2** Extraction kit efficiency. RNA extraction from AccuPlex viral particles was examined using direct PCR (black column) and compared to four different combinations of storage liquids and extraction kits including Maxwell RSC viral extraction kit with water (red circles), ethanol (EtOH; blue squares), and DRS (green inverted triangles), as well as Zymo Quick-DNA/RNA viral kit with DRS (turquoise diamonds), followed by quantification using RT-qPCR assay. Values are expressed as nucleocapsid (N1) copy numbers in 5 $\mu$l of RNA extract; all replicates are plotted as individual points, with means presented as columns. Direct PCR was treated as 100% to calculate the extraction efficiency of the other extraction methods (recorded within the columns). Significant differences were determined by Welch's $t$ test, and significance ($P < 0.05$) is denoted by asterisks.

efficiency of the automated process, the AccuPlex viral particles were directly added to 96-well PCR plates and subjected to thermal/enzymatic treatments before performing RT-qPCR. This direct PCR method was considered 100% (average 327 copies) and compared with other methods employed during this study. The comparative extraction efficiencies of the automated system with H$_2$O, ethanol (EtOH), and DRS were 61.0%, 61.5%, and 55.9%, respectively, while the manual method with DRS was 62.2%. All the extraction procedures exhibited high variabilities; however, the automated system demonstrated a lower coefficient of variation (4.0 to 7.7%) compared to manual kit extraction (8.4%).

**E2E assay.** ZeptoMetrix NATtrol, an inactivated SARS-CoV-2-positive control, was used in these studies since the AccuPlex stock contains high concentrations of glycerol, making it challenging to dry onto material surfaces. For this study, 5,000 copies of NATtrol viral particles per 25 cm$^2$ were spotted on bare stainless steel (BSS), painted stainless steel (PSS), polyethylene terephthalate modified with glycol (PETG), and fiberglass-reinforced plastic (FRP) materials. After desiccation of the viral control on the surface, the viral droplets left a visible plaque on all surfaces (smaller dots within the swabbed area, Fig. 3A). Sample collection with the swab showed noticeable differences in the amount of the plaque that was dissociated during swabbing. The visible marks associated with BSS, PSS, and FRP materials remained mostly intact after swabbing; however, roughly half of the PETG plaques broke apart during swabbing (Fig. 3A). Such plaque breakup possibly allowed for collecting larger pieces of the viral plaques on the PETG. Materials of 25 cm$^2$ (coupons) were swabbed 18 h after inoculation (day 1) and reswabbed (with a fresh swab) after incubating at room temperature for additional 24 h (day 2). On day 8, after inoculation, the coupons were wiped down with 0.6% bleach (sodium hypochlorite) and then reswabbed with a fresh swab for the third time.

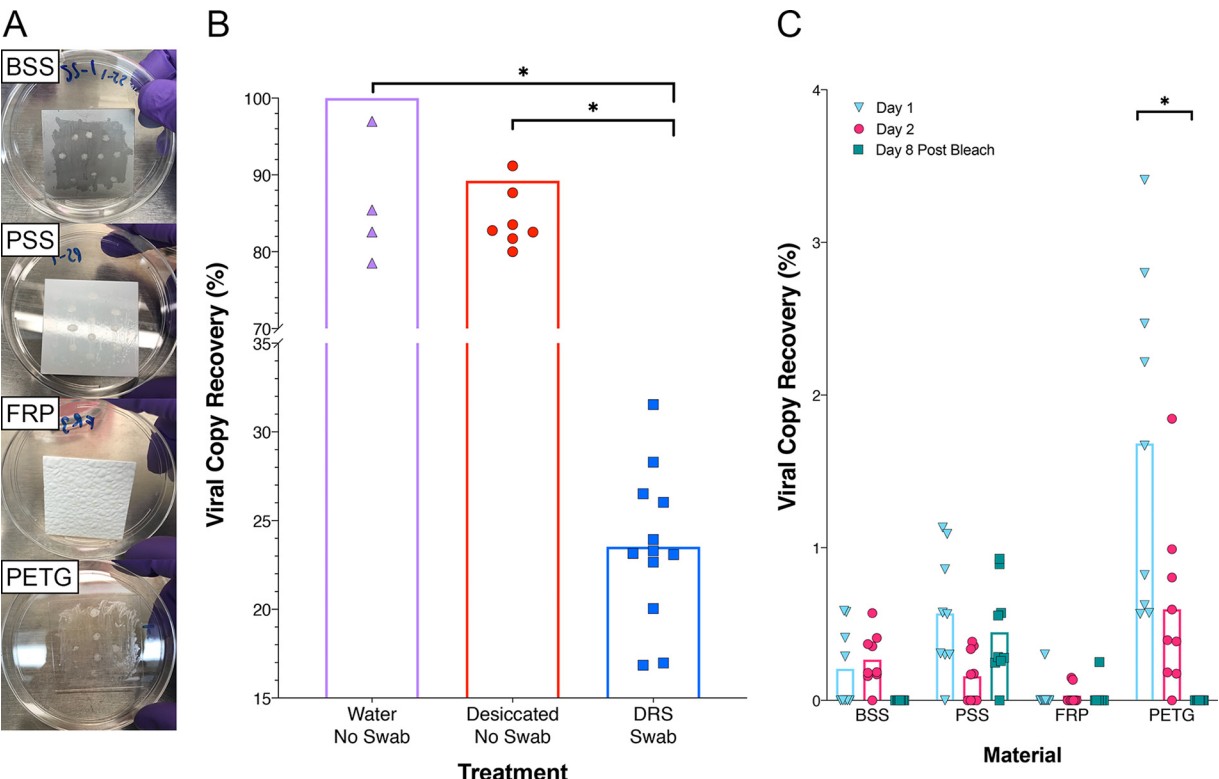

**FIG 3** Viral particle recovery from built environment surface materials. (A) Image of the inoculated plates (BSS, PSS, FRP, and PETG) where inactivated viral particles (10 $\mu$l of NATtrol) were aliquoted 10 times onto four separate materials in triplicate. (B) Viral particles were either kept overnight at room temperature as liquid (Eppendorf tube was closed; Water No Swab, purple triangles) or desiccated overnight at room temperature in tubes (No Swab Desiccated, red circles) which were then sacrificed to extract RNA directly without removing from the surface. In addition, aliquots of viral particles that were not desiccated but inoculated in DRS and swab materials were also processed (DRS Swab, blue squares). (C) Viral particles were collected from the seeded surfaces with Isohelix swabs and DRS, extracted on the Maxwell RSC, and quantified using RT-qPCR assay. Viral RNA copy number for each condition was divided by an extraction control to calculate percent recovery for day 1 (blue inverted triangles), day 2 (red circles), and day 8 post-bleach (green squares). Statistical significance was determined by Welch's $t$ test with significance ($P < 0.05$) denoted by asterisks.

Initially, when viral particles were desiccated in an Eppendorf tube and RNA was extracted directly, an 11% loss of RNA due to desiccation was documented compared to the solution that was not dried. The average percent recovery of the viral particles directly inoculated onto the swabs in DRS was ~23% (Fig. 3B). However, when desiccated on materials, the highest percentage of RNA recovery after day 1 was observed for PETG material (1.68%), followed by PSS (0.57%) and BSS (0.21%) (Fig. 3C). The lowest observed recovery was from FRP at 0.03% (Fig. 3C). On day 2, viral recovery decreased on PETG (0.6%) and PSS (0.16%) coupons; however, no decrease was noted on day 2 for the BSS and FRP materials (Fig. 3C). After treatment with 0.6% bleach, the recovery from BSS and PETG materials decreased to below detection limit (BDL), while the bleach was not effective in the removal of RNA from PSS as traces of RNA could still be detected (0.45% recovery), while only 0.03% recovery was observed on FRP. This might be because the RT-qPCR assay could detect very short fragments of RNA (~70 bp) and hence likely amplified degraded nucleic acids. This test revealed that viral persistence on surfaces varies, and in some cases (such as on PSS) viral RNA can be recovered after cleaning with bleach.

**Comparison of RT-LAMP and RT-qPCR assays.** Reverse transcription loop-mediated isothermal amplification (RT-LAMP) is a single-step colorimetric presence-absence assay that can be used as a narrow-range semi-quantitative assay to determine the presence of SARS-CoV-2. When used as described by the manufacturer, the color-imetric RT-LAMP assay generates a yellow color for a positive result or remains unchanged (pink) for a negative result. Samples with borderline results have a gradient

color change between yellow and pink. All samples can be further analyzed to obtain narrow-range semi-quantitative results by measuring the resulting DNA product with the Qubit DNA broad-range quantification kit. Since the dynamic range between positive and negative is narrow, negative reactions (pink) have final DNA concentrations of <50 ng/$\mu$l post amplification, and full-color positives have a post amplification concentration of ~550 ng/$\mu$l in a 25-$\mu$l reaction mixture. Samples that are borderline will have DNA yields between 50 and 550. For these studies, both positive viral controls were lysed directly (see Table S1 in the supplemental material), and the lowest limit of detection was determined at 5 and 12.5 copies per reaction mixture for AccuPlex and NATtrol, respectively (Fig. 4A). Since the RT-LAMP assay is not truly quantitative, values of <150 ng $\mu$l$^{-1}$ DNA concentration were considered negative, and values of ≥150 ng $\mu$l$^{-1}$ were considered positive.

The samples collected from the inoculated coupon (Fig. 3A) and analyzed by RT-qPCR were further analyzed using the RT-LAMP assay (Fig. 4B). When the RT-qPCR results were compared with the RT-LAMP assay results, >300 copies were definitively positive with the RT-LAMP assay (yellow coloration). However, for samples with concentrations near the limit of detection (LOD) for RT-LAMP assay (~12.5 copies/$\mu$l), the results between RT-qPCR and RT-LAMP were less correlative (Fig. 4B). For samples that exhibited discrepancies between the two assays (BSS2 and PETG3), Sanger sequencing was performed and confirmed SARS-CoV-2 sequences (Fig. 4C), indicating the reliability of the RT-LAMP assay.

**Material-associated organic inhibition in the RNA recovery.** Since many of the chemicals associated with cleaning, disinfection, and indigenous chemical constituents of the materials could have PCR inhibitors, we conducted several experiments to determine this potential. The precision-cleaned uninoculated surface materials (25 cm²) described above (BSS, PSS, PETG, and FRP) were swabbed and placed in DRS medium along with 5,000 copies of the NATtrol viral control. These samples were processed along with a positive control that included a swab in DRS medium with NATtrol viral control but not exposed to any test surfaces. Results indicated that all swabs used for sample collection of surface materials demonstrated similar recovery rates as the controls not used in surface sampling. Analysis of variance (ANOVA) indicated that swabs used to sample both BSS and FRP had similar recovery rates of 25.2% and 24.3%, respectively, while PSS had 30.8% and PETG had 36.0% (Fig. S1). Swabs sampled from PETG had a significantly higher recovery rate than FRP ($P = 0.0001$) and BSS ($P = 0.0006$), while PSS was significantly higher than FRP ($P = 0.0152$) and BSS ($P = 0.0439$) surface types. The recovery percentages exhibited for all swabs were within a standard deviation (average 29.5% ± 5.5% standard deviation). These recovery rates from various tested materials (24% to 36%) were similar to those of the positive control that was not exposed to any test surfaces (20% to 25%; Fig. 3B), which demonstrated that the precision cleaning did not leave residual organics or debris that could inhibit the RT-qPCR assay. The difference in the recovery was attributed solely to the DRS-swab combination, since viral particles spiked in water with swab without DRS solution demonstrated an ~88% recovery.

**Influence of environmental debris on viral quantification.** To determine whether the detection of SARS-CoV-2 virus would be affected by the debris associated with built environment materials (stainless-steel metal, Amerstat, plastic, copper, painted surfaces, and wood), samples were collected and efficiency of the E2E procedure was tested. The collected materials in DRS solution were inoculated with and without AccuPlex (500 copies) viral standards prior to RNA extraction and RT-qPCR assay. As expected, there was a significant inhibition in the recovery of the AccuPlex viral RNA from all surface materials swabbed, ranging from 50% recovery for stainless steel to only 4% and 8% for Amerstat and the painted surfaces, respectively (Fig. 5). Wood, copper, and plastic surfaces exhibited intermediate recovery of viral particles at a level of ~20%. On average, the recovery of AccuPlex viral RNA from the stainless steel was significantly higher than that from all other tested surfaces ($P = 0.05$ to 0.004), but the

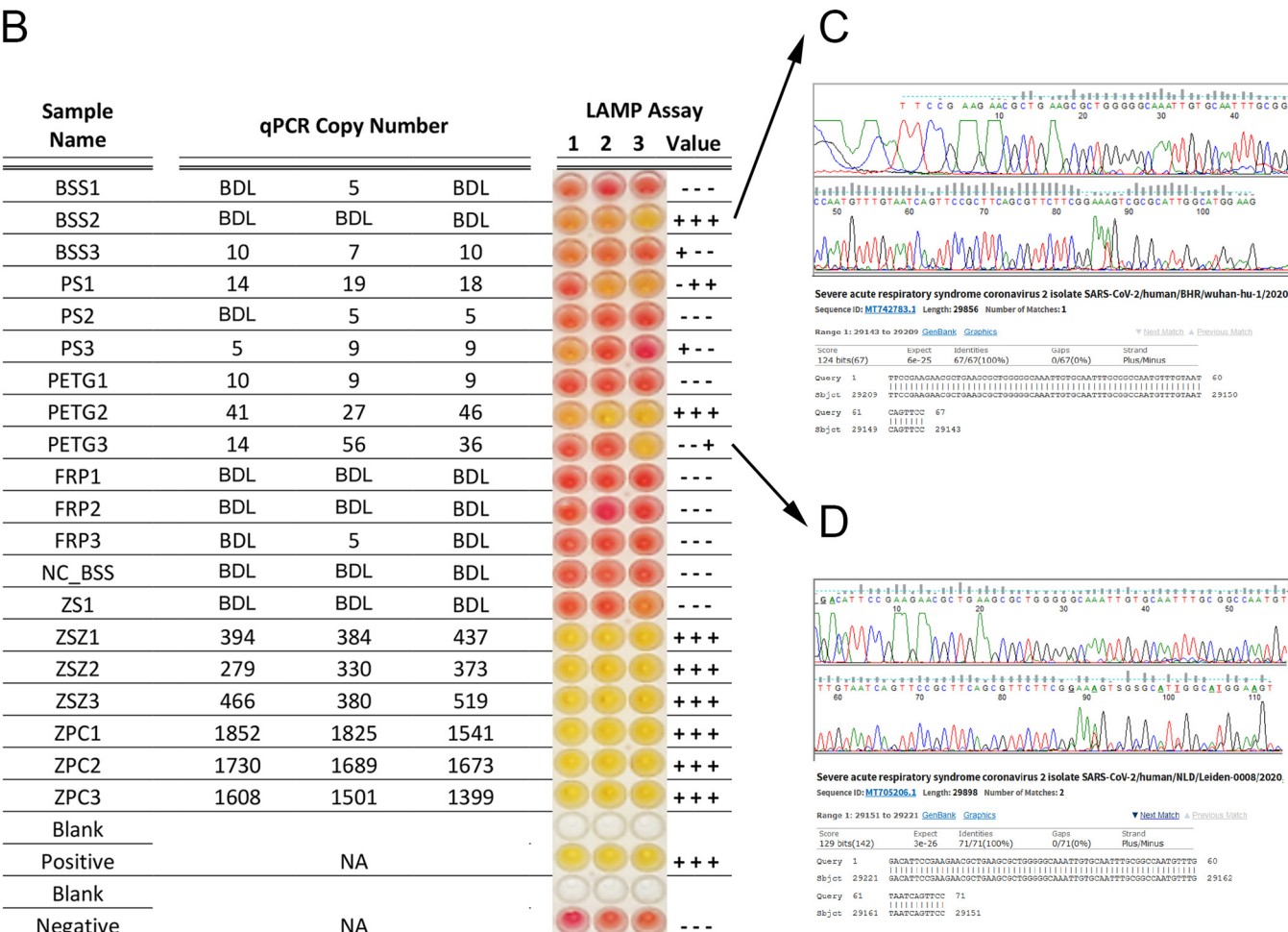

**FIG 4** Comparison of RT-LAMP and RT-qPCR assays. (A) RT-LAMP assay for limit of detection was carried out for AccuPlex and NATrol standards with both the colorimetric changes seen in the reaction (RT-LAMP assay output) and the Qubit quantifications presented across a dilution series of viral particle number. The qualitative RT-LAMP assay output was determined based on color change from red to yellow in the presence of the target sequence, whereas RNA measurements of RT-LAMP assay reactions using Qubit gave semi-quantitative values. Qubit values that were below 150 ng/μl were denoted as minus signs, and Qubit values that were above 150 ng/μl were recorded as plus signs. (B) Viral particles collected from built environment surface materials (Fig. 3, day 1) were analyzed with the RT-LAMP and RT-qPCR assays. RT-LAMP assay colorimetric output is presented alongside Qubit +/− result value and RT-qPCR quantities. Values that were not tested were marked as not applicable (NA), and values that were undetectable were recorded as BDL. A BSS coupon that remained uninoculated (NC_BSS) and was processed alongside as a negative control; a swab negative control in DRS (ZS1); a swab with 5,000 copies of NATtrol in DRS (ZSZ1-ZSZ3); 5,000 copies of NATtrol control extracted directly from Maxwell (ZPC1-ZPC3). Sanger sequence chromatograms and associated National Center for Biotechnology Information BLAST results for BSS2 (C) and PETG3 (D) are also included.

AccuPlex RNA recovery was more variable with a range from 15% to 100%. Barring one or two outliers, the recovery from plastics was consistent (Fig. 5).

To ascertain whether the decreased AccuPlex viral RNA recovery was due to the interaction of the environmental debris with the organics in extraction reagents, the post-extracts of the six materials were spiked with 500 copies of synthetic fragments obtained from Integrated DNA Technologies (IDT) prior to being subjected to RT-qPCR.

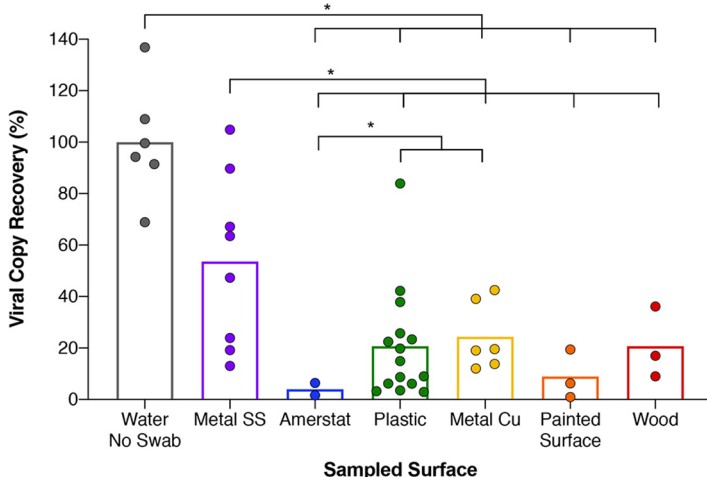

**FIG 5** Inhibition by field-collected built environment surface samples after RNA extraction. Field swab collection of diverse built environment surface samples which had their DRS vials spiked with inactivated viral AccuPlex particles prior to RNA extraction and quantification with RT-qPCR. Differential amplification of stainless steel (Metal SS) (purple circles), Amerstat (blue circles), plastic (green circles), copper (Metal Cu) (yellow circles), painted surface (orange circles), and wood (red circles) was compared to a positive control (gray circles) and reported as percent recovery compared to that positive-control mean. Each column represents average percent recovery for the respective surface type. Significance ($P < 0.05$) is denoted by asterisks, based on Welch's $t$ test.

In contrast to the results above, which showed considerable inhibition, amplification of the spiked synthetic IDT fragments after the RNA extraction was largely unaffected. The copper resulted in the lowest recovery at 77%, followed by plastic, wood, and painted surfaces (~84%), while the stainless steel and the Amerstat exhibited 90% recovery compared to the control (Fig. S2). These results underscore how the type of environmental surface can influence the recovery of viral molecules, while the RNA purification kit chemistry might account for a small percentage of such inhibition.

**Validation of E2E process by an independent laboratory.** The E2E assay was repeated using the same materials by an independent laboratory for reproducibility and verification purposes. The independent evaluation included LOD determination of RT-qPCR assay, RNA extraction efficiency of automated system/manual kits, and recovery of NATtrol viral particles from various built environment material surfaces. The results of the second laboratory evaluation were comparable and/or equivalent to the results presented here. A standalone report is included in Data Set S1.

**Built environment study testing SARS-CoV-2 from environmental surfaces.** The E2E protocol developed during this study could confirm viral presence from built environment surfaces only when ≥1,000 viral particles per 25 cm² were present due to the losses associated with swab collection, transportation solution, RNA extraction, and material surface retention. Despite these limitations, the combination of using an Isohelix swab, DRS as transportation medium, automated RNA extraction, and RT-qPCR assay was determined to be the best available E2E protocol during March 2020 to reproducibly detect and measure SARS-CoV-2 from built environment surfaces. The E2E process implemented during this study is shown in Fig. 6. The samples collected were from seven different materials found in 10 buildings, including stainless steel, Amerstat, plastic, copper, and painted surfaces. All selected surfaces were in areas of the facilities with large amounts of pedestrian traffic and deemed high-touch surfaces capable of serving as SARS-CoV-2 fomites. None of the 368 samples collected tested positive for SARS-CoV-2 (i.e., RT-qPCR amplification for N1 gene was BDL) using the E2E process developed during this study. Since the detection sensitivity of the E2E process implemented was 1,000 viral particles per 25 cm², the samples collected from built environmental surfaces were either devoid of the targeted virus or BDL of the E2E assay.

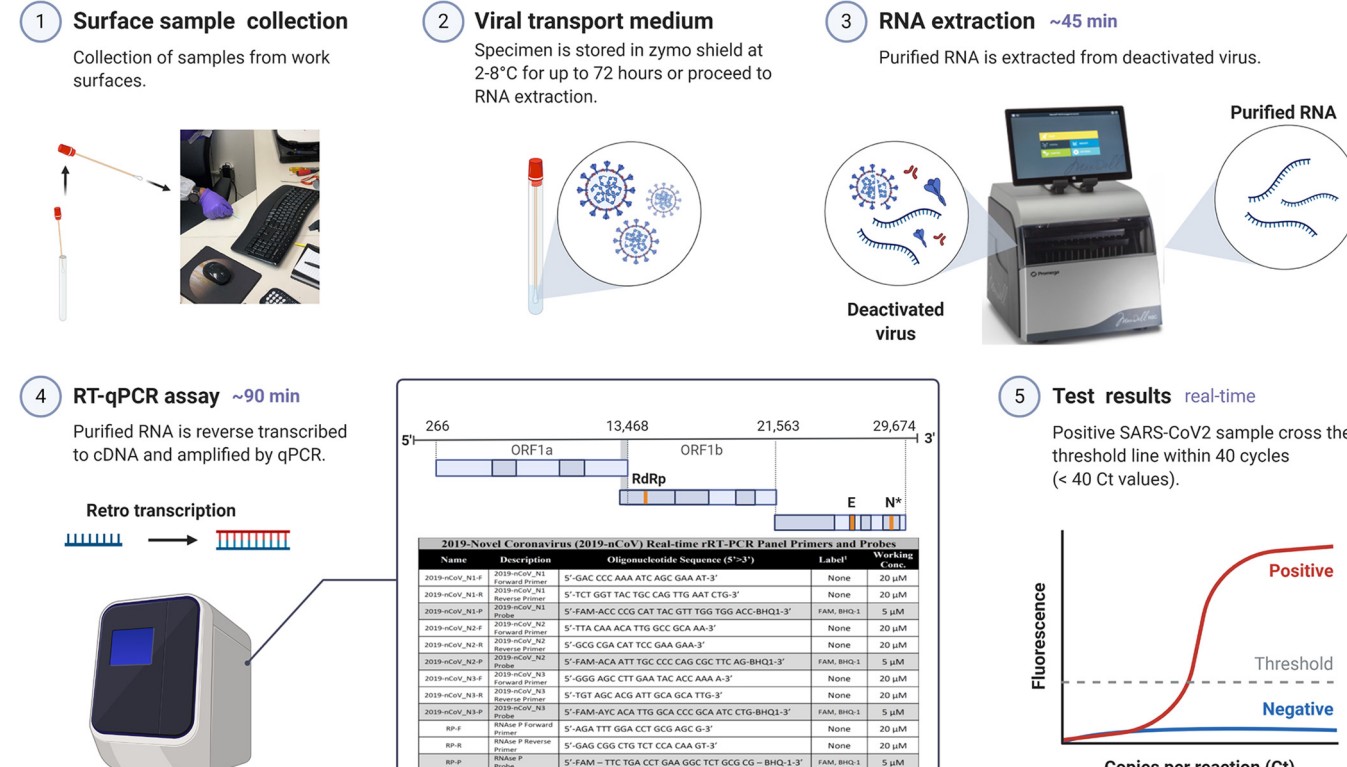

**FIG 6** Environmental surface testing using E2E protocol. The optimized E2E protocol for detecting SARS-CoV-2 virus on surfaces is a 5-part procedure: (1) surface sample collection, (2) viral transport medium, (3) RNA extraction, (4) RT-qPCR assay, and (5) test results.

## DISCUSSION

The current clinical method for screening potential SARS-CoV-2 virus patients requires an initial throat and/or nasopharyngeal swab sample collection (9). Unlike clinical samples, fomites and high-touch surfaces that become contaminated with the virus display lower concentrations of the virus (10), which are often difficult to detect due to method limitations and, in some cases, inhibitory materials. For this reason, robust methods are imperative for the recovery and detection of SARS-CoV-2 from environmental surfaces. Previous studies have analyzed a variety of methods for viral recovery from surfaces (11); however, there are a substantial number of variables that can impact collection, processing, and quantification of viral particles. Despite the World Health Organization's "how-to" guide for SARS-CoV-2 surface sampling in hospital settings, this is the first comprehensive study to adequately address all of the issues associated with an E2E assay for surface sampling and detection of SARS-CoV-2.

During this study, Isohelix swabs were selected over Copan swabs due to easier handling and higher sensitivity for sample collection, and this approach has been successfully used by other studies (12). Furthermore, our results demonstrated that automated RNA extraction was as efficient (13, 14) as manual kits for extracting SARS-CoV-2, which has been previously noted using phenol-chloroform (15).

At the outset of this study, in February 2020, there were several molecular methods available for assaying the virus in a given sample. Various reports demonstrated well-established techniques, such as RT-qPCR (16), RT-LAMP (1), poly(A) transcriptome sequencing (RNA-seq) (17), ribodepletion RNA-seq and methylated RNA immunoprecipitation sequencing (MeRIP-seq) (18), direct RNA sequencing (19), capture panel/amplicon (20), and digital droplet PCR (ddPCR) (21). While each of these techniques has its strengths and merits, in the context of a diverse, low-biomass sample, each has its own shortcomings, which limits its use for environmental surveillance applications. This includes elements such as detection limits, costs, inhibitor effects, input volumes, result

type, or ability to validate a positive result. However, despite these shortcomings, two distinct molecular technologies (RT-qPCR and RT-LAMP assay) evolved to become part of the mainstream research toolkit for both clinical and environmental testing (22). The benefits of these assays, when run in tandem, help resolve data associated with the more challenging and complex environmental sample types to accurately detect and quantify SARS-CoV-2 from various surface materials.

In comparing RT-qPCR, RT-LAMP, and ddPCR to detect SARS-CoV-2 from surface samples, ddPCR was found to be the most accurate and repeatable diagnostic tool. However, ddPCR is more expensive and requires a specialized ddPCR instrument. In contrast, the RT-LAMP assay is faster and less labor-intensive than ddPCR, is relatively inexpensive, and requires minimal instrumentation to operate (e.g., a heat block, water bath, or thermocycler). The results of the RT-LAMP assay are colorimetric, and the low infrastructure requirements make the assay ideal for field testing as demonstrated in past studies (23, 24). Since RT-LAMP is a narrow-range semi-quantitative assay, accurate quantification is best performed by RT-qPCR, the gold standard for viral RNA detection (25–27). For our studies, we selected RT-qPCR for its wide dynamic range, throughput, and sensitivity (2 copies/$\mu$l) as the primary analysis tool, with RT-LAMP as the confirmatory assay.

Once an infected person begins shedding SARS-CoV-2 viral particles, the primary route of infection is via respiration, through either droplets or aerosols that are expelled during normal speech, respiration, and especially sneezing and unintentionally inhaled by even healthy individuals (28). Although the virus appears to be primarily transmitted through air, SARS-CoV-2 can remain viable on surfaces for up to 72 h (6, 7). Thus, similar to other respiratory viruses (29), it is likely that a major route of SARS-CoV-2 infection comes from contact with infected surfaces followed by inadvertent touching of the face and mouth. This pattern was observed in a study in Wenzhou, China, where numerous individuals became infected despite not having any direct contact with known patients (30). These findings, in combination with virus longevity on surfaces, strongly suggest that transmission is not limited to just aerosols. Additionally, preliminary research suggests that the infective dose is lower for SARS-CoV-2 than for other respiratory infections (31, 32). These findings highlight the importance of effective environmental surveillance, surface monitoring, and proper sanitization methods to eliminate the virus.

In this study, we analyzed several environmental surface materials that were inoculated with a known concentration of SARS-CoV-2 viral reference standard to determine the recovery efficiencies for each material. While each material has characteristics that contribute to recovery, surface roughness and hydrophobicity are important contributors. It has been reported that surface roughness is a key mitigating factor for lower recovery of biological materials (33). For the test surfaces evaluated in this study, FRP had a textured surface and resulted in a lower NATtrol viral recovery. Even though the PETG surface had similar surface roughness as BSS and PSS, the surface texture of PETG might have enabled the recovery of more virions. This was evident from visual observations revealing that the dried virus inoculum was easily dissociated and resulted in higher recoveries—a likely result of both the smoothness and hydrophobicity.

The decreasing amount of recoverable NATtrol viral particles over time from the surface materials is likely attributable to the combination of desiccation time (day 1 to 8) and the use of a disinfectant. Unlike all other materials, PSS demonstrated nearly the same RNA copy numbers persisting from day 1 desiccation until day 8 post-bleach. This might be due to the RT-qPCR method, which was targeting only 67- to 72-bp amplicons that could still easily be detected from virion fragments degraded by the disinfectant. Despite desiccation, the chemistry of the paint associated with PSS material might have allowed viral fragments to persist even after cleaning with bleach. Furthermore, after applying bleach, the pigmentation of the paint was altered, indicating that a chemical reaction had occurred, which may have enabled easier removal of viral particles from the surfaces. Similarly, recovery of viral fragments was documented on the cruise ship *Diamond* after hypochlorite disinfection of contaminated rooms (34). These surprising

positive results were likely due to the degraded RNA fragments still being detected in the short-amplicon RT-qPCR method. Due to short-fragment amplification, even significantly degraded RNA can be detected. In order to avoid these false-positive results after the bleach treatments, samples should be tested using an alternative technique that targets longer RNA fragments, such as RT-LAMP. In addition, nucleic acid intercalating dyes were reported to be useful in eliminating naked nucleic acids and compromised microbial structures for bacteria (35), fungi (36), and DNA/RNA viruses (37).

To detect the high concentration of SARS-CoV-2 virus in clinical samples, it was shown that direct amplification was possible with a maximum sample input of 2 $\mu$l (38), whereas for environmental samples, an RNA purification step was mandatory due to the PCR-inhibitory substances (13, 39). In addition, the concentration of the target molecules during RNA extraction allowed larger volume input (10-fold more) and increased the detection limit (2 copies per $\mu$l of RNA extract). The collection of microorganisms from environmental surfaces has been documented to yield ~1% to 10% of biological materials due to issues in their removal from the surfaces as well as challenges associated with their dissociation from the swab (13, 39). During our study, the DRS chemistry in combination with environmental debris and RNA extraction has compounded losses an additional ~80%.

The E2E process implemented to survey SARS-CoV-2 virus presence for built environment surfaces ($n = 368$ samples) exhibited no viral incidence (or <1,000 viral particles per 25 cm²), which might be attributable to highly controlled safety practices that were strictly followed. These practices, including (but not limited to) admitting a limited number of employees at a given time period, "Safe at Work" training, enforcing social distancing, wearing masks, practicing personal hygiene, and deep-cleaning of the environmental surfaces, might have limited the viral contamination in these built environment surfaces. However, high-traffic areas like hospitals, restaurants, cruise ships, and subways might show a different pattern(s) of viral adherence and persistence on fomites and surfaces (34, 40).

**Conclusion.** When examining all elements, the optimized E2E protocol implemented during this study indicated that only ~0.5% to 2% of the viral particles could be recovered from a variety of built environment surfaces and a minimum of 1,000 target molecules (viruses) per 25 cm² was needed to positively detect the virus. During this study, it was established that 1% of NATtrol viral particles were recovered due to sample collection (swabs) and transportation solution (DRS) and that the RNA extraction step accounted for a further 90% loss of target molecules. These data reflect an overall E2E process efficiency of 0.1%, meaning that at least 1,000 copies need to be present for successful and reproducible detection of the SARS-CoV-2 virus from environmental surfaces.

## MATERIALS AND METHODS

**Inactivated viral reference standards.** Two noninfectious, replication-deficient, encapsulated SARS-CoV-2 viral reference standards were used during this study, including the SeraCare AccuPlex (Milford, MA; catalog no. 0505-0126), which contained the ORF1a, RdRp, E, and N sequences, and the ZeptoMetrix NATtrol [Buffalo, NY; catalog no. NATSARS(COV2)-ERC], which contained the entire RNA sequence. The AccuPlex and NATtrol stocks were purchased at a concentration of $5 \times 10^3$ and $5 \times 10^4$ viral particles per ml, respectively. These concentrations were confirmed in-house using digital droplet PCR (ddPCR) to be within $5 \pm 1.2$ copies for AccuPlex and $48.3 \pm 3.2$ copies for NATtrol viral standards (see Table S2 in the supplemental material).

Digital droplet PCR was performed using the Bio-Rad QX200 instrument with the IDT primer/probe set for N1 and N2 with a modified probe quencher of Iowa Black ZEN/IBFQ (catalog no. 10006770) along with the Bio-Rad one-step RT ddPCR advanced supermix (catalog no. 1864021). Four methods were used for extraction of RNA from these reference materials (i.e., AccuPlex and NATtrol) and consisted of the following: (a) direct lysis at 75°C for 5 min, (b) direct lysis of a 1:1 mixture of sample to nuclease-free water (15 $\mu$l:15 $\mu$l) to which 3 $\mu$l of proteinase K (Qiagen) and 0.8 $\mu$l of RNase inhibitor (Ribolock; Thermo Scientific EO0381) were added and which was incubated at 50°C for 10 min followed by freeze-thaw of −80°C to +95°C for 4 min, (c) utilization of viral RNA extraction kits such as the QIAamp viral RNA minikit (Qiagen, Germantown, MD; catalog no. 52904), and (d) the RNeasy Micro kit (Qiagen; catalog no. 74004). Volumes of 1, 2, 3, 5, and 7 $\mu$l of the AccuPlex and NATtrol viral standards were analyzed using methods 1 and 2 for ddPCR to determine exact copy number (Table S2).

**Swab and viral transport medium selection.** Two protocols, involving sample collection and transport medium, were tested during this study. The first protocol, in Fig. S3A, shows the procedure used for the Metagenomics and Metadesign of Subways and Urban Biomes (MetaSUB) and heritage NASA environmental sampling (12, 13, 41). The Copan liquid Amies elution swab (ESwab; Copan Diagnostics; catalog no. 480C) was used for environmental sampling. Sampled Copan swabs were stored on dry ice and transferred to the lab for further processing. Once in the lab, 300 $\mu$l of lysis buffer and 30 $\mu$l proteinase K (Promega, Madison, WI) were added, and the swab was cut using sterile scissors to release the swab into the tube and mixed thoroughly using a vortex. The materials released from the swab were extracted using the Maxwell viral total nucleic acid purification kit (AS1330; Promega) or the Zymo Quick-DNA/RNA viral kit (catalog no. D7020). The protocol shown in Fig. S3B represents the reference protocol procedures used for a similar study design by the 2017–2019 MetaSUB research consortium (12). This process used the Isohelix MS-02 swab (miniswab, Isohelix catalog no. MS-02) with 400 $\mu$l of DNA/RNA Shield (DRS; Zymo Corp.; R1100-250) preservative. Sampled Isohelix swabs were broken off into the sample tube and transferred to the lab at room temperature. Once in the lab, samples were extracted for nucleic acids via the Promega Maxwell RSC 16. Among the swabs tested, Isohelix swabs demonstrated a higher recovery of microorganisms than that of the Copan swabs (12). Since no published reports were available on the efficiency of swabs specific for virus collection from environmental surfaces, data from the MetaSUB consortium (12) were adapted for this study. The DRS medium used throughout this study contains proprietary chemicals that inactivate the live virus and preserve RNA at a biosafety level 2 status.

**Efficiency of various protocols in extracting viral RNA.** The standard methodology for viral RNA extraction in this study involved using the surface samples collected in DRS ($\sim$200 $\mu$l) and processing them using the automated Maxwell RSC extraction platform (Promega Corp., Madison, WI) following the manufacturer's instructions for the Maxwell RSC viral total nucleic acid purification kit (Promega). In brief, the collected swabs were vortexed for 2 min and treated with the lysis solution provided by the manufacturer (220 $\mu$l of the lysis buffer per 100 $\mu$l of sample and 200 $\mu$l of the DRS solution). This extraction tube was incubated at room temperature for 10 min and 56°C for an additional 10 min. Samples were transferred to Maxwell cartridges for extraction using the viral total nucleic acid program of the instrument. Purified RNA was eluted into 60 $\mu$l of UltraPure molecular-grade water and divided into two aliquots. Samples were stored at −80°C with one aliquot being used for downstream RT-qPCR analysis while the other aliquot was archived for later use.

In order to compare the efficiencies of the extraction protocols and the effects of the DNA/RNA Shield on RNA amplification, four sets of extraction fluids were prepared in triplicate. Set 1 was prepared with 100 $\mu$l of AccuPlex in 100 $\mu$l of UltraPure water, set 2 was prepared with 100 $\mu$l of AccuPlex in 100 $\mu$l 95% EtOH, and sets 3 and 4 were prepared with 100 $\mu$l of AccuPlex in DRS. Sets 1, 2, and 3 were all processed on the Maxwell RSC as described above. Set 4 was processed using the Quick-DNA/RNA viral kit (Zymo Research, Irvine, CA) following the manufacturer's protocol. Purified RNA was eluted in 60 $\mu$l of UltraPure water.

**Synthetic RNA and limit of detection for RT-qPCR.** Two synthetic nucleic acid reference samples were used to generate standard curves for the RT-qPCRs: (i) 2019-nCoV N positive DNA control (10006625) from Integrated DNA Technologies (IDT) (42) and (ii) SARS-CoV-2 RNA control 2 (MN908947.3) from Twist Biosciences (San Francisco, CA). The IDT standard consisted of control plasmids containing the complete nucleocapsid gene from SARS-CoV-2, while the Twist standard consisted of six synthetic 5-kb single-stranded RNA (ssRNA) sections of the viral genome. Both IDT and Twist contain the nucleocapsid gene and can be amplified by either N1 or N2 primer sets, producing amplified products that have lengths of 72 bp or 67 bp, respectively. Comparison of N1 and N2 primers using the IDT and Twist BioSciences synthetic standards showed that all combinations of the primers and standards had highly reproducible amplification quantities across log dilutions, with N1 demonstrating a slightly higher efficiency amplification curve (Fig. S4A to E). Hence, only the N1 primer set was used for developing the E2E protocol. Samples that resulted in N1-positive results were further confirmed with the N2 primer set. A significantly higher viral copy number was detected using the IDT reference material (1.28-fold) in comparison to Twist ($P < 0.05$) when assessed with RNA extracted from AccuPlex as a benchmark control (Fig. S4E).

To determine the limits of detection of the RT-qPCR assay, a 2-fold dilution series from 0 to 200 viral RNA copies per reaction volume (5 $\mu$l; 12 replicates) was conducted and indicated a LOD of 10 viral RNA copies per 5 $\mu$l reaction volume (2 copies/$\mu$l; Fig. S5). Among the 12 replicates that theoretically contained one viral RNA copy (5 copies/5 $\mu$l), five did not reach the cycle threshold ($C_T$) and were thus considered BDL. All no-template controls (NTCs) were negative. As expected, the standard deviation of $C_T$ values increased as the molecule concentration decreased ($<10$ copies) (Fig. S5, inset).

**Optimization of RT-qPCR assay.** qPCR was carried out with the extracted viral RNA from the sample using the Luna universal probe one-step RT-qPCR kit (catalog no. E3006; New England BioLabs [NEB], USA) per the manufacturer's protocol for Applied Biosystems real-time instruments. N1 and N2 IDT primers (2019-CoV CDC EUA kit, Integrated DNA Technologies) designed for CDC SARS-CoV-2 qPCR probe assays were used for all reaction setups. The kit consists of all published SARS-CoV-2 assays in the CDC's recommended working concentration. The final 20-$\mu$l reaction mix also included Antarctic thermolabile UDG (uracil-DNA glycosylase) to prevent sample cross-contamination. The IDT SARS-CoV-2 plasmid DNA control was used to generate a $\log_{10}$ standard curve from 1 to $10^5$ copies in triplicate. The AccuPlex SARS-CoV-2 reference material was used as an extraction control and treated as an "unknown" sample for each analysis. A QuantStudio 6 Flex real-time PCR detection system was used for all RT-qPCR runs. Cycling conditions were as follows: reverse transcription, 55°C (10 min, 1×); initial denaturation,

95°C (1 min, 1×); and 40 cycles of 95°C (15 s), 60°C (60 s) plus plate read. The N1 gene was used to determine the number of viral particles in a sample. NTCs, a reaction mixture with molecular-grade water substituted for the sample, were run on each RT-qPCR plate to serve as negative controls. Standard curve and quantification were carried out using the Design and Analysis software version 2.4.1, for QuantStudio 6/7 Pro systems.

There are some unresolved issues with the RT-qPCR, which repeatedly detected larger amounts of AccuPlex and NATtrol in standard controls than the quantity that was being measured by dd-qPCR. Figure S6A and B shows the number of copies per ml of AccuPlex and NATtrol that were calculated based on RT-qPCR runs. The red lines demarcate the reported concentration of viral particles for AccuPlex and NATrol. Figure S6C shows that there was a 2.69-fold-higher concentration of viral particles for AccuPlex and a 3.5-fold-higher concentration for NATtrol.

**RT-LAMP assay.** A 5-$\mu$l aliquot of each sample was analyzed in triplicate using the RT-LAMP assay with the WarmStart RT-LAMP reagent (M1800S; NEB Inc., Ipswich, MA) and the N2/E primer mix against the nucleocapsid envelope protein gene. A custom primer mix for the final primer mix included 40 mM guanidine hydrochloride, which increased the sensitivity as previously described (43). All samples were incubated at 65°C for 42 min and photographed. Sample were quantified using spectrofluorimetry with the Qubit broad-range DNA kit (ThermoFisher, Waltham, MA). Titrations were performed on both AccuPlex and NATtrol viral particles for estimating copy numbers. Direct RNA extraction was performed by mixing viral reference particles at a ratio of 1:1 (15 $\mu$l to 15 $\mu$l of water) and adding 1 $\mu$l of an RNase inhibitor (RiboLock; ThermoFisher; EO0381) and 3 $\mu$l of proteinase K (Qiagen, Germantown, MD), with incubation at 50°C for 10 min followed by immediate freezing at −80°C. After freezing, the controls were immediately incubated at 95°C for 4 min, followed by duplicate titrations into the RT-LAMP reaction master mix and incubation at 65°C for 42 min.

**Surface materials tested and coupon fabrication.** Four of the most common high-touch surface materials were used in this study, including bare 302 stainless steel (BSS), painted 302 stainless steel (PSS; white acrylic paint 168130 [Rust-Oleum, Vernon Hills, IL]), polyethylene terephthalate modified with glycol (PETG), and fiberglass-reinforced plastic (FRP). All materials were smooth on a macroscale, except for FRP, which exhibited an irregular, textured surface. These materials were fabricated as test "coupons" of 25 cm² square at the Jet Propulsion Laboratory (JPL) and sand tumbled to deburr. Throughout, coupons were handled carefully so as to limit surface damage and scratching.

**Precision cleaning of the test coupons.** Unless otherwise indicated, all precision cleaning was performed in a Class 100 biohazard hood or a Class 100 laminar flow bench. Care was taken in handling, and high-grade chemicals were used to minimize contamination. Coupons were precision cleaned based on each individual material's best practices, as outlined in JPL's standard protocols (44). In short, BSS was cleaned per JPL D-51981 type IV (subsequent baths of solvent, detergent [AquaVantage 815 GD; Brulin Holding Company, Indianapolis, IN], and alkaline and final passivation). The passivation consisted of a 30-min exposure to 5 M nitric acid at 24°C. Due to the paint's associated chemical attributes and susceptibility to solvents, PSS was rinsed with deionized water. The FRP wood laminate and PETG were both cleaned per the JPL D-51981 type V method C (solvent bath followed by deionized water rinse). After cleaning, the product cleanliness level was tested to the level of 100 (which means particles of <0.5 $\mu$m do not exceed 100 particle counts). The cleaned test coupons were individually sealed in an antistatic Amerstat bag until use.

**Inoculation of surface materials (test coupons).** Precision-cleaned coupons were opened aseptically in a biosafety cabinet and placed into individual, sterile petri dishes. Aliquots of 10 $\mu$l of the NATtrol control were spotted ($n = 10$) onto each test coupon in evenly spaced rows of 3, 4, and 3 spots and covered with a lid. Triplicate coupons of each of the following material types were prepared, including several controls: (i) a BSS coupon that remained uninoculated (NC BSS) and was processed alongside as a negative control; (ii) a swab negative control in DRS (ZS); (iii) a swab with 5,000 copies of NATtrol in DRS (ZSZ); and (iv) 5,000 copies of NATtrol control extracted directly from Maxwell (ZPC). All test coupons were loaded into a modified GasPak system, anaerobic jar 150 LG (catalog no. 260607; Becton, Dickinson, Franklin Lakes, NJ) with the palladium catalyst removed and a valve port drilled into the top of the lid; no desiccation beads or reactants were added. The lid port was hooked up to a vacuum line to provide negative pressure on the jar. Coupons were dried at room temperature for 18 h, sampled, and immediately extracted for RNA (day 1; initial collection). After initial swabbing, coupons were stored at room temperature for 24 h at standard pressure and swabbed again with a fresh swab, followed by viral RNA extraction (day 2; secondary collection). Coupons were subsequently stored at room temperature and standard pressure for another 5 days, after which coupons were treated with 10% bleach (0.6% [vol/vol] sodium hypochlorite) using Kimwipes (Kimberly-Clark, Irving, TX) using a wiping pattern vertically, horizontally, and diagonally. Coupons were allowed to dry for 30 min, swabbed, and extracted for RNA (day 8; collection after bleach treatment).

**Sample collection from coupons or built environmental surfaces.** Test coupons were sampled over a 25-cm² area using Isohelix MS-02 buccal swabs (Cell Projects, Kent, UK). Prefilled 2-ml tubes containing 200 $\mu$l of DRS and labeled with a unique barcode (catalog no. R1100-96-1) were used for each sample. The Isohelix swab was dipped into DRS solution for 15 s prior to sampling to ensure the swab was sufficiently moistened. The moistened swab was then held against the sample surface at a 45-degree angle and dragged in a raster pattern across the 25-cm² area. To ensure good coverage over the sample area, the raster pattern was repeated three times in different directions (horizontal, vertical, and diagonal), rotating the swab head 180 degrees between the horizontal and vertical passes. The swab was held to the surface perpendicular to the direction of travel to ensure that the maximal surface area was covered by the swab during each pass in the raster pattern. After sample collection, each swab head

was transferred into the same barcoded tube used to premoisten the swab by aseptically breaking and twisting the head off into the tube. Sample-specific metadata (e.g., surface type and finish) were recorded for each barcoded tube. Environmental sampling of built environment surfaces was conducted in an identical manner, except that the moistened swab for the field control was not touched to a surface but rather was waved in the air for 2 min prior to breaking off the swab head into a barcoded DRS tube. After collection of all samples, DRS collection tubes were stored at room temperature for up to 3 h before RNA extraction.

Environmental sampling was conducted only in buildings that were actively being utilized and that had personnel. All sampling was performed in these buildings in the morning prior to the routine daily cleaning of each building to ensure that any SARS-CoV-2 virions remaining on the surfaces from the previous workday would not be removed prior to our sampling. Each building varied in the number of active staff members who passed through on a daily basis. Unfortunately, due to confidentiality concerns and agreements with facility managers, exact numbers cannot be given; however, we can confirm that each building had anywhere from 1 employee to tens of employees working across the 10 buildings sampled in this study. Additionally, due to confidentiality concerns it is not possible to relate whether there were any COVID-19-positive staff members (or family) identified in the buildings during the time of the sampling. Personal protective equipment, including masks and in some cases visors/goggles, were strictly enforced on the working employees along with frequent handwashing and the sanitization of surfaces after their use. In addition, in-house "Safe at Work" training to follow strict guidelines was provided to the employees.

**Environmental debris in the recovery of viral particle/RNA.** Various materials, including metal, Amerstat, plastic, wood, copper plate, and painted surface, were tested from 10 separate facilities to assess whether the debris associated with the environmental surface affected the recovery and detection of the viral RNA. Each surface was sampled with two swabs, which were preserved in DRS. The DRS from both collection tubes corresponding to one type of the environmental surface was pooled, mixed, and divided into two 200-$\mu$l aliquots. One aliquot was inoculated with 100 $\mu$l of AccuPlex control and the other with 100 $\mu$l of UltraPure water, and they were extracted using the Maxwell RSC using the protocol for the RT-qPCR analysis. The percent recovery for each tested material was determined compared to the control containing 100 $\mu$l of AccuPlex SARS-CoV-2 in 200 $\mu$l of UltraPure water. Three technical RT-qPCR replicates of biological samples were used for the analysis.

**Inhibition of Maxwell extraction chemistry in the recovery of viral particle/RNA.** Test surfaces were evaluated to determine if any RNA extraction inhibition was observed for the AccuPlex viral particles using the Maxwell RSC system. In order to determine the potential effect of the Maxwell RSC kit, 500 copies of the IDT synthetic fragments (2019-CoV N-positive control; Integrated DNA Technologies, Inc., San Jose, CA) were added to each test reaction and 5 $\mu$l of each extracted sample. Twenty microliters of the Luna master mix (catalog no. E3006; NEB, Ipswich, MA) was added, and the plate was sealed and analyzed using the QuantStudio 6F instrument. The percentage of the inhibition for each tested material was determined by comparison to the control containing 500 copies. Three technical RT-qPCR replicates of biological samples were used for the analysis.

**Material-associated debris and chemistry inhibition.** In order to determine whether debris or chemistry associated with the built environment surface materials contained PCR inhibitors, uninoculated test coupons were swabbed following the standard swabbing procedure outlined above. The swabs were transferred to a tube containing DRS and 100 $\mu$l of the NATtrol viral particles. Appropriate negative and positive controls were also included. Samples were then extracted for RNA using the Maxwell RSC and were analyzed via RT-qPCR. Inhibition was determined by comparison of the extraction ratios between positive control and the sample reaction mixtures.

**Development of SWAB metadata generation.** For the field data collection and associated metadata characteristics of samples, the JPL Information and Technology Solutions Directorate created a custom mobile application (Safe Workspace Analysis Barcode Scanner [SWABS]) that uses an iPhone to capture tracking metadata at each stage of the sample collection and analysis process. The vendor-provided barcode of each sample container was chosen as the unique identifier of each sample tracked through each phase of analysis. In the first phase (sampling), the container barcode was scanned and relevant metadata such as the name of the collection personnel, location, surface material properties, time of collection, material lot numbers, and optional description details were recorded for each sample. To accelerate data entry and eliminate human errors, barcode scanning with the iPhone camera was used to exactly identify each sample, and data in common were retained and automatically reused. In the second phase (RNA extraction), the sample was scanned once again, and details of the Maxwell machine identifier, extraction tip size, and lot numbers were added. For the third phase (archival), analysts added details of the cryobox location, extraction tip size, and lot numbers for each sample. At the final qPCR stage, the record was completed with the qPCR machine identifier, extraction tip size, and lot numbers, as well as the $C_T$ score and copy number determined by the analysis. At each stage of the analysis procedure, the operations procedure version number was also recorded to track which documented procedure was followed at the time when each sample was processed. All of these sample processing data were gathered and stored in a centralized database at JPL. Leveraging this database, the SWABS web application makes these data accessible for viewing, searching, or editing, as well as providing reporting capabilities to communicate and summarize any of the data on demand. Although the SWABS application streamlines and improves the accuracy of the processing metadata-recording process as a whole, it is most advantageous during the initial sample collection phase owing to its mobile platform that allows its users to move around freely within the workspace environment while minimally encumbered by support equipment.

**Statistical analyses.** All statistical analyses were performed using GraphPad Prism version 8.2.0 (GraphPad Software, San Diego, CA, USA). Specifically, Welch's *t* test and a two-way ANOVA followed by a *post hoc* sample correction were computed. Outliers were screened using the rOut method from the robustX R package (https://CRAN.R-project.org/package=robustX).

## SUPPLEMENTAL MATERIAL

Supplemental material is available online only.

**FIG S1**, PDF file, 0.03 MB.
**FIG S2**, PDF file, 0.04 MB.
**FIG S3**, PDF file, 0.5 MB.
**FIG S4**, PDF file, 0.3 MB.
**FIG S5**, PDF file, 1.2 MB.
**FIG S6**, PDF file, 0.3 MB.
**TABLE S1**, PDF file, 0.4 MB.
**TABLE S2**, PDF file, 0.1 MB.
**DATA SET S1**, PDF file, 7 MB.

## ACKNOWLEDGMENTS

Part of the research described in this publication was carried out at the Jet Propulsion Laboratory, California Institute of Technology, under a contract with NASA. Researchers associated with Biotechnology and Planetary Protection Group at JPL are acknowledged for their facility support.

We thank Garry Burdick and Subbarao Surampudi (initiating the concept of COVID-19 surface testing); Soren Madsen (day-to-day management of the workflow); Roger Gibbs, Leon Alkalai, Timothy O'Donnell, and the JPL Management Council (financial support and directions); Mimi Ton (IRB clearance); Anton Ovcharenko (safety protocol development); Kerry Wisden, Marisa Gamboa, and Bill Kert (procuring chemicals and materials); and Oscar Rendon Perez (coupon material preparation and precision cleaning). We also thank Brent Mcwatters, Mark Powell, and the members of the JPL Information and Technology Solutions Directorate who rapidly created an iPhone app that was used for metadata collection during this project. We are indebted to the personnel involved in JPL shipping/receiving for their help during this pandemic and the facility managers associated with the surface collection locations. Additionally, we thank Mikael Kubista from the TATAA Biocenter for helping to develop our RT-qPCR limit of detection protocols, and we give thanks to MetaSUB consortium members (especially Benjamin Young) for generating the sample collection SOP and Julie Dragon for RT-LAMP assay SOP and continued support. Christopher Fleming (Promega) and Frank Tansley (Thermo Fisher Scientific) are acknowledged for their timely support of procuring critical equipment and consumables. We thank Tara Ellison, technical support scientist from Bio-Rad, for assisting on ddPCR, and Dan Butler from Cornell Medicine as well as Nathan Tanner from NEB for collaborating on the RT-LAMP analysis during the very early parts of this study. David Lee from JPL is thanked for critically reviewing the manuscript.

This research was supported by the JPL Director Discretionary Funds for COVID-19 projects which also funded a portion of the fellowship of C.W.P., A.B., and J.M.W. We also thank Igor Tulchinsky and the WorldQuant Foundation, Bill Ackman and Olivia Flatto and the Pershing Square Foundation, Ken Griffin and Citadel, Testing for America (501c3), OpenCovidScreen Foundation, the Bert L and N Kuggie Vallee Foundation, the U.S. National Institutes of Health (R01AI125416, R21AI129851, R01AI151059), and the Alfred P. Sloan Foundation (G-2015-13964). The funders had no role in study design, data collection and interpretation, the writing of the manuscript, or the decision to submit the work for publication.

This manuscript was prepared as an account of work sponsored by NASA, an agency of the US Government. The US Government, NASA, California Institute of Technology, Jet Propulsion Laboratory, and their employees make no warranty, expressed or implied, or assume any liability or responsibility for the accuracy, completeness, or usefulness of information, apparatus, product, or process disclosed in this manuscript,

or represents that its use would not infringe upon privately held rights. The use of, and references to any commercial product, process, or service does not necessarily constitute or imply endorsement, recommendation, or favoring by the US Government, NASA, California Institute of Technology, or Jet Propulsion Laboratory. Views and opinions presented herein by the authors of this manuscript do not necessarily reflect those of the US Government, NASA, California Institute of Technology, or Jet Propulsion Laboratory, and shall not be used for advertisements or product endorsements.

Work for this study way also completed by Biotia, Inc. Biotia and its employees make no warranty, expressed or implied, or assume any liability or responsibility for the accuracy, completeness, or usefulness of information, apparatus, product, or process disclosed in this manuscript, or represents that its use would not infringe upon privately held rights.

K.V. coordinated with all authors in designing the concept, executed the study, implemented the project, was involved in the data analyses, and wrote the manuscript. C.W.P. was involved in establishing the RT-qPCR assay with N.S., was instrumental in executing the E2E process, and wrote the manuscript along with K.V. N.S. carried out the QA/QC of the RT-qPCR assay and analyzed the data. S.T. helped design and set up the experimental methods; provided training; conducted RT-LAMP, qPCR, ddPCR, and interpretation of data; and assisted in writing and editing the manuscript. P.L. carried out lab work pertaining to protocol optimization of RT-LAMP, qPCR, and ddPCR assays as well as Sanger sequencing. N.S., J.M.W., C.W.P., R.H., and K.C. carried out environmental sampling from various built environment surfaces. J.M.W. managed metadata collection and curation, helped analyze data, and was crucial in surface sampling as well as drafting portions of the manuscript. A.B. performed RNA extraction of the collected samples, contributed to data analysis and interpretation, and drafted parts of the manuscript. C.U. helped analyze the data and write the paper and assisted in the study design. R.H. helped with sample collection and contributed in writing about the swab and DNA extraction methodology process selection for the paper. K.C. planned and coordinated built environment sampling and supported data collection during sampling. A.S. and P.V. carried out RT-qPCR assays and contributed to the LOD determination. B.G.C. oversaw the project, provided critical input for the QC/QA analyses of RT-qPCR data, and edited the manuscript. N.B.O. coordinated work at Biotia, where a second laboratory performed verification, and helped design the additional procedures used. M.C.-R. helped design additional experiments, performed experiments, interpreted and analyzed results, and contributed to writing the validation portion of Data Set S1. D.B. conducted experiments pertaining to the selection of swabs and DRS, wrote those portions of the manuscript, and reviewed the manuscript. C.E.M. coordinated with K.V., helped to design the concept, and provided insights about the MetaSUB protocol to get it implemented.

All authors who participated in this study have reviewed the results, read the final manuscript, and given their consent for publication.

N.B.O., M.C.-R., and C.E.M. hold shares in Biotia, a company that conducts infectious disease diagnostics and characterization. All other authors declare that they have no competing interests.

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
