## [Reviewer comments · mSystems]

End-to-End Protocol for the Detection of SARS-CoV-2 from Built Environments

Ceth Parker, Nitin Singh, Scott Tighe, Adriana Blachowicz, Jason Wood, Arman Seuylemezian, Parag Vaishampayan, Camilla Urbaniak, Ryan Hendrickson, Pheobe Laaguiby, Kevin Clark, Brian Clement, Niamh O'Hara, Mara Couto-Rodriguez, Daniela Bezdán, Chris Mason, and Kasthuri Venkateswaran

Corresponding Author(s): Kasthuri Venkateswaran, California Institute of Technology

Review Timeline:

Submission Date:	August 10, 2020
Editorial Decision:	September 8, 2020
Revision Received:	September 16, 2020
Accepted:	September 22, 2020

Editor: Sean Gibbons

Reviewer(s): Disclosure of reviewer identity is with reference to reviewer comments included in decision letter(s). The following individuals involved in review of your submission have agreed to reveal their identity: Fuqing Wu (Reviewer #1)

Transaction Report:

DOI: <https://doi.org/10.1128/mSystems.00771-20>

September 8, 2020

Dr. Kasthuri Venkateswaran
California Institute of Technology
Jet Propulsion Laboratory
Mail Stop 245-105
4800, Oak Grove Dr.
Pasadena, CA 91109

Re: mSystems00771-20 (End-to-End Protocol for the Detection of SARS-CoV-2 from Built Environments)

Dear Dr. Kasthuri Venkateswaran:

Below you will find the comments of two reviewers. As you can see, both reviewers appreciated the timeliness and value of your work. The reviewers had several comments and questions that should help you to improve the quality and clarity of this manuscript. Please address all reviewer concerns and provide a point-by-point response, along with a revised version of your manuscript.

To submit your modified manuscript, log onto the eJP submission site at <https://msystems.msubmit.net/cgi-bin/main.plex>. If you cannot remember your password, click the "Can't remember your password?" link and follow the instructions on the screen. Go to Author Tasks and click the appropriate manuscript title to begin the resubmission process. The information that you entered when you first submitted the paper will be displayed. Please update the information as necessary. Provide (1) point-by-point responses to the issues raised by the reviewers as file type "Response to Reviewers," not in your cover letter, and (2) a PDF file that indicates the changes from the original submission (by highlighting or underlining the changes) as file type "Marked Up Manuscript - For Review Only."

Due to the SARS-CoV-2 pandemic, our typical 60 day deadline for revisions will not be applied. I hope that you will be able to submit a revised manuscript soon, but want to reassure you that the journal will be flexible in terms of timing, particularly if experimental revisions are needed. When you are ready to resubmit, please know that our staff and Editors are working remotely and handling submissions without delay. If you do not wish to modify the manuscript and prefer to submit it to another journal, please notify me of your decision immediately so that the manuscript may be formally withdrawn from consideration by mSystems.

Sincerely,

Sean Gibbons

Editor, mSystems

Journals Department
Reviewer comments: (see attachment)

Reviewer #2 (Comments for the Author):

This manuscript describes the systematic examination of the efficiencies of procedures from sampling to qPCR and the impact of each individual step on the recovery of SARS-CoV-2 from surfaces. The authors conclude by providing a practical protocol for surface monitoring of SARS-CoV-2 with its recovery rate and detection limit, which greatly aids the interpretation of results. The information presented in this study, particularly the granular breakdown of losses at each step, could also be valuable to the detection of other pathogens from the built environment. The study is overall thorough, and the results are of immediate interest.

A few specific comments:

It appears that the number of data points collected for different conditions (water vs. DRS, different surface samples) varied widely. Please comment on the effect of this imbalance on the statistical analysis. If the variation reflects exclusion of certain datapoints, please explain the exclusion criteria.

For longer term impact, considering providing some details on the proprietary swab types and kits used (e.g., materials or important properties of the kit), such that the overall findings might still be useful after the manufacturers change their design or become defunct.

I. 69 Specify which government.

II. 86-88 A monitoring "plan" might include details like which surfaces should be sampled and how frequently. More accurately, this study outlines a comprehensive method that can be used in environmental monitoring. Please rephrase.

I. 115 Over how much time? Are results likely to vary over time? Same for temperature.

I. 319 Intriguing hypothesis. Why not measure roughness and hydrophobicity here?

I. 333 This is an important point that bears emphasizing. Please comment on the potential implications of RNA-based detection methods for assessing disinfection, which largely targets membrane/capsid integrity. Perhaps also comment on the potential for additional assessment of integrity (e.g., <https://www.medrxiv.org/content/10.1101/2020.05.08.20095364v1>).

II. 325-326 What is meant by the chemical nature of the paint? Please elaborate.

II. 545-547 Rephrase: "Field control samples were collected in a similar manner to environmental samples, but instead..."

I. 551 What type of debris? Where did these samples come from?

I. 571 This section needs more detail. Again, where did samples come from? Define what controls were included.

Typos and other errors: l.73, l. 78, l. 282

In this paper, Parker et al developed an end-to-end protocol to detect SARS-CoV-2 on four different surface materials in built environment, and systematically investigated the recovery efficiency for each step using surrogate viruses. With this protocol, they found SARS-CoV-2 surrogate could be detected even after 8 days on two material surfaces after bleach treatment. They also determined the limit of detection for the protocol, which is 1000 viral particles per 25 cm². Finally, they collected 368 samples from seven different material surfaces in 10 buildings, and found all of them were negative of SARS-CoV-2.

One nice point is that the authors tried to develop an automated protocol for SARS-CoV-2 detection in built environments, especially the custom mobile program to collect field data and associated metadata. This can be more standardized and scaled up when necessary, and points to the application of this E2E protocol in environmental studies.

My major comment refers to the surrogate virus they employed to test the recovery efficiency. It is unclear that what kind of viral particles they used, but their surface structure may be very different from the real SARS-CoV-2 virions. This difference could significantly impact the viral RNA stability in the environment, and thus the main conclusion about SARS-CoV-2 persistence on surfaces. The authors tested about 400 real environmental samples, and the results were all negative, which is good except that it is hard to conclude the protocol's efficiency, but I defer to the editor about this point.

I also have some specific comments for the author:

1. Line 78: there is a typo.
2. Lines 110-111: since there are basically DRS and Swab two variables, may be better to specify the "various combinations".
3. Why the efficiency for swab and DRS combination is so low? DRS medium could inactivate the virus and avoid degradation, which is supposed to be better than Water. What is the material for the swab, hydrophobic or hydrophilic? SARS-CoV-2 is hydrophobic, maybe that property could explain the significant viral particles sequestration in the swab. It is apparently Water is better than DRS to resuspend the sample, why the authors keep using the DRS in the following experiments?
4. Fig 1: what do the dots and bar represent? Mean of replicates or results from N1/N2 primers? The dots variation looks pretty large (40% -95% and 5%-45%). The data showed the relative recovery of viral copy, it would be nice to also have an absolute quantification of recovery rate for the water-no swab group, this is important to evaluate the whole procedures' efficiency.
5. Lines 372-372: "These concentrations were confirmed in-house using digital

- droplet PCR (ddPCR) to be within 1.25% accurate (Supplemental Table 1).” The 1.25% estimate looks incorrect based on the ddPCR results (47 and 5.89 copies/ul from Table S1).
6. Lines 374-376: missing experimental details about the ddPCR, what is the total droplet number for each sample after emulsion? This number is important for determining the accuracy of viral concentrations. How the authors determine the base line? What is the variation range for the each sample? Whether the confidence levels for samples overlap with Control sample?
 7. Lines 383-384: it seems the authors tried different volumes (1,2,3,5,7 ul) of viral standards for ddPCR, however Table S1 only showed 2ul to 3ul. As to Table S1, the first and second row for the “Treatment” column is the same, they are replicates or a typo (one may be N2 primer)?
 8. Line 155, the highest recovery rate is from PETG material, is that because of the extra modification of glycol to the PET material and decreases the sample’s dry process.
 9. Result for the Bleach treatment on the four materials did not show.
 10. Lines 185-186: the inconsistent results between RT-qPCR and RT-LAMP are associated with the materials? Maybe different inhibitions from the materials.
 11. Figure 4B, what are the ZS1, ZSZ1, ZPC? For sample BSS2, all replicates are negative from qPCR result, but LAMP results are all positive. Is that because of contamination?
 12. The authors tried to repeat the detection with LAMP assay, however, it is unclear what conclusions they were trying to make. It looks like the LAMP has a worse sensitivity than qPCR, even though in some other reports people claim the LOD is ~2 copies per ul of samples. I can hardly see how this section is related to the main topic of this paper.
 13. Lines 211-219: Did the author test the RNA yield after the extraction? The reason why the authors saw lower recovery for samples containing environmental debris is probably because of the low RNA yield for those samples. And the low-yield is not resulted from “inhibition”, but just because of those debris can physically impact the column’s RNA binding capability during the extraction. It is a typical step to remove large cell debris or visible solid materials in the supernatant before transferring lysed samples onto the column during RNA extraction procedure. However, since the authors were using an automated extraction machine, it didn’t “notice” this issue.

14. Lines 243: It is unclear why the authors claim the limit is 1000 viral particles per 25 cm². They only tested 5000 copies in previous sections, and get positive results for some measurements.
15. Lines 245-247: why the authors think this combination with DRS medium is the best? In figure 1, figure 2, and figure 3, the samples resuspended in Water are consistently better than in DRS medium (100% vs 75%).
16. Lines 248-252: 368 samples were collected from seven different material surfaces in 10 buildings, and found all of them were negative of SARS-CoV-2. Whether they were people working in the building during the sample collection period? How many people in and out of the building per day? Whether there was a positive covid19 patients (even related) found in the buildings? Whether people in the building were required to wear masks? All those information are necessary in order to interpret the Negative results.
17. The authors ran technical replicates for qPCR, what if one replicate is positive within the range of quantification and the other is negative? How was this handled?
18. There is no Supplemental Figure 5A or 5B. Does the authors re-check the original positive controls' concentration before determining the limit of detection?

**Answers for Reviewer-1:**

In this paper, Parker et al developed an end-to-end protocol to detect SARS-CoV-2 on four
different surface materials in built environment, and systematically investigated the recovery
efficiency for each step using surrogate viruses. With this protocol, they found SARS-CoV-2
surrogate could be detected even after 8 days on two material surfaces after bleach treatment.
They also determined the limit of detection for the protocol, which is 1000 viral particles per 25
7 cm². Finally, they collected 368 samples from seven different material surfaces in 10 buildings,
and found all of them were negative of SARS-CoV-2.

One nice point is that the authors tried to develop an automated protocol for SARS- CoV-2
detection in built environments, especially the custom mobile program to collect field data and
associated metadata. This can be more standardized and scaled up when necessary, and points to
the application of this E2E protocol in environmental studies.

My major comment refers to the surrogate virus they employed to test the recovery efficiency. It
is unclear that what kind of viral particles they used, but their surface structure may be very
different from the real SARS-CoV-2 virions. This difference could significantly impact the viral
RNA stability in the environment, and thus the main conclusion about SARS-CoV-2 persistence
on surfaces. The authors tested about 400 real environmental samples, and the results were all
negative, which is good except that it is hard to conclude the protocol's efficiency, but I defer to
the editor about this point.

**Ans:** The authors agree that the type of viral particle plays a significant role in the viral
recovery from a surface, and that modifications to the viral capsid and other structural
features may lead to drastically altered recovery outcomes. Thus, the authors spent
significant time identifying the closest test analog to SARS-CoV-2 since the use of live virus
was prohibited by our Institutional Review Board (IRB). However, these commercially
available virus particles are compatible with assays targeting CDC and WHO consensus
sequences. At the time of the initiation of this study there were two inactivated SARS-CoV-2
virus materials available to develop various protocols. We tested commercial products from
both Seracare and Zeptomatrix. As requested by the reviewer, their surface structure analysis
were not carried out by the authors; however, the commercial vendors (SeraCare) confirmed
that these viral particles are “highly stable, fully commutable, recombinant virus with fully

intact viral particle” as shown in the figure above. In addition, “Zeptomatrix NATrol
products are ready to use, inactivated full process controls designed to evaluate performance
of molecular tests. They can be used for verification of assays, training of laboratory
personnel and to monitor assay-kit lot performance. Furthermore, NATrol products contain
intact virus and should be run in a manner similar to clinical specimens.”

I also have some specific comments for the author:

1. Line 78: there is a typo.

Ans: The authors have corrected the typo to read,
“between individuals via fomites, compromising the ability”.

2. Lines 110-111: since there are basically DRS and Swab two variables, may be better to
specify the “various combinations”.

Ans: The authors have modified the text to read (Line #111 to 113),

“The resulting viral copy numbers were then compared and computed to understand the
effects of swabs and DRS solution individually, along with the combined impact of
swabs and DRS, in the recovery of viral particles (Figure 1).”

3. Why the efficiency for swab and DRS combination is so low? DRS medium could inactivate
the virus and avoid degradation, which is supposed to be better than Water. What is the
material for the swab, hydrophobic or hydrophilic? SARS- CoV-2 is hydrophobic, maybe
that property could explain the significant viral particles sequestration in the swab. It is
apparently Water is better than DRS to resuspend the sample, why the authors keep using the
DRS in the following experiments?

Ans: As pointed out by the reviewer, water seems to be better than any transportation
medium; however, unlike water, DRS medium can inactivate the virus and prevents RNA
degradation. Due to the pandemic, the IRB enforced the institution to deactivate the samples
prior to analysis. A variety of deactivation methods were tested by the authors and compared
with those found in the literature (including ethanol, Hydrogen Peroxide, Sodium
Hypochlorite, and DRS medium [proprietary preservative]) leading to the selection of DRS
medium. The authors found that DRS medium both deactivated the virus and served as a
transport medium, maintaining viral genetic integrity for long periods of time, a necessity
when working with RNA viruses. The MetaSUB consortium (Danko et al., 2019)
recommended use of DRS medium to transport samples from various parts of the world
clarifying that the biological materials (bacteria/fungi/virus) in DRS medium were
inactivated without losing the integrity of the genetic materials (Supplementary Figure S3B).

As pointed out by the reviewer, a swab head with either hydrophobic or hydrophilic
characteristics could jeopardize the ability to collect samples, and then release them when
back into solution. The Isohelix swabs are designed to retain liquid (not like polyester), while
not absorbing large quantities of moisture (not like cotton) and hence the swabs can be
wetted but not retain the materials collected onto the swab. This can’t be characterized as
either hydrophilic or hydrophobic.

4. Fig 1: what do the dots and bar represent? Mean of replicates or results from N1/N2 primers?
The dots variation looks pretty large (40% -95% and 5%-45%). The data showed the relative
recovery of viral copy, it would be nice to also have an absolute quantification of recovery

rate for the water-no swab group, this is important to evaluate the whole procedures'
efficiency.

Ans: The dots represent technical replicates for the N1 primers while the bar represents mean
of N1 primer replicates. As mentioned earlier in this review forum, at the beginning of this
study in early March 2020, there were not many commercially available inactivated SARS-
CoV-2 viral particles on the market. Our data on SeraCare AccuPlex viral standards using the
Zeata View (Particle Metrix, Meerbusch, Germany) electrophoresis and Brownian motion
video analysis laser scattering microscopy technique demonstrated that Accuplex was
composed of a high viscosity suspension buffer in glycerol with large amounts of clumped
particles and debris. Similarly, Zeptomatrix NATtrol viral standard also contained unwanted
particles that were used to stabilize the virus. The absolute quantitation was performed for
both AccuPlex as well as Zeptomatrix viral standards using ddPCR assay. PCR reactions
were set up by diluting 1 to 7 μ L of viral standard in 25 μ L reaction mixtures. The resulting
ddPCR counts were used as 100% which agreed with the vendor provided counts. The
absolute copy numbers as measured by ddPCR were 5 ± 1.2 copies for AccuPlex and $48.3 \pm$
3.2 copies for NATtrol viral standards (*Supplemental Table 2*).

- 5. Lines 372-372: "These concentrations were confirmed in-house using digital droplet PCR
(ddPCR) to be within 1.25% accurate (Supplemental Table 1)." The 1.25% estimate looks
incorrect based on the ddPCR results (47 and 5.89 copies/ μ l from Table S1).

This was inadvertently placed as % accuracy but this is standard deviation. The text will now
read as (See Line# 378 to 380):

"These concentrations were confirmed in-house using digital droplet PCR (ddPCR) to be
within 5 ± 1.2 copies for AccuPlex and 48.3 ± 3.2 copies for NATtrol viral standards
(*Supplemental Table 2*)."

6. Lines 374-376: missing experimental details about the ddPCR, what is the total droplet
number for each sample after emulsion? This number is important for determining the
accuracy of viral concentrations. How the authors determine the base line? What is the
variation range for each sample? Whether the confidence levels for samples overlap with
Control sample?

The ddPCR experimental details are provided in Line #381 to 392. We ran several trials. The
baseline was no template controls since the input was directly lysed samples and the
concentration of the controls were very low. Accuplex is 5 copies per μ l and Zepto NATtrol
viral standard was 50 per μ l. Therefore, we could use exactly 1, 2, or 3 μ l directly to the
master mix. Since these concentrations were very low, they were run in absolute
concentration mode compared to the no template control (NTC) as baseline and the NTC was
required to be below detection limit. These assays were set up in collaboration with the
BioRad ddPCR technical team. An example of one run is given in a new *Supplemental Table*
*2*.

About the variation we added the following sentences in Line #378 to 380: These
concentrations were confirmed in-house using digital droplet PCR (ddPCR) to be within 5
± 1.2 copies for AccuPlex and 48.3 ± 3.2 copies for NATtrol viral standards (*Supplemental*
*Table 2*).

When ~20K droplets analyzed in ddPCR, we did not see any false positive for the negative
control.

7. Lines 383-384: it seems the authors tried different volumes (1,2,3,5,7 ul) of viral standards
for ddPCR, however Table S1 only showed 2ul to 3ul. As to Table S1, the first and second
row for the “Treatment” column is the same, they are replicates or a typo (one may be N2
primer?)?

Ans: We tested volumes of viral particles from 1 – 7 μ L as stated in the manuscript at various
stages; however, Table S1 shows representative results (2, 2.5, and 3 μ L). Data were not
shown for the other input volumes. A footnote in Table S1 clarified this discrepancy. Two
treatments named “75°C 5 Min -No ProK, No Freeze, No RNase inhibitor-N2” were the
same and the modified footnote says as follows:

“Digital droplet qPCR was performed using the BioRad QX200 instrument, the IDT
primer/probe set for N1 and N2 with a modified probe quencher of Iowa Black -
ZEN/IBFQ (Cat# 10006770). The BioRad One-Step RT ddPCR advanced supermix
(1864021) was used as the master mix for ddPCR. Samples were directly lysed using
either a direct lysis of 75°C for 5 min for the SeroCare Accuplex or a Proteinase k/ freeze
thaw –80°C to +95°C 4 min for the Zeptometrix standard. Treated samples were analyzed
directly by ddPCR in triplicate at 1, 2, 2.5, 3, 5, and 7 μ l direct input to the reaction. Here
we present representative results for 2, 2.5, and 3 μ L”. Two treatments named “75°C 5
Min -No ProK, No Freeze, No RNase inhibitor-N2” were the same but run on two
different days (time separate time points).

8. Line 155, the highest recovery rate is from PETG material, is that because of the extra
modification of glycol to the PET material and decreases the sample’s dry process.

Ans: One of the materials selected for this study was PETG because of its prevalent use in
built environments, and was to serve as one of two representative plastic surfaces (along with
FRP). Hence the differential drying process between PET and PETG were not tested.

9. Result for the Bleach treatment on the four materials did not show.

Ans: The authors plotted the recovery from all the bleach treatment of coupons (see Figure
3C green squares). However, most of the samples were below detection level so they are
plotted as 0.

10. Lines 185-186: the inconsistent results between RT-qPCR and RT-LAMP are associated with
the materials? Maybe different inhibitions from the materials.

Ans: As suggested by the reviewer, inhibition of surface material/debris were also suspected
by the authors, hence, we tested this hypothesis by artificially inoculating synthetic RNA
fragments after nucleic acid extraction. The results of this experiment are depicted in
Supplemental Figure 2 which confirmed the removal of inhibitor substances by the RNA
extraction system employed. However, the inconsistent results between RT-qPCR and RT-
LAMP were mainly due to the limit of detection.

11. Figure 4B, what are the ZS1, ZSZ1, ZPC? For sample BSS2, all replicates are negative from
qPCR result, but LAMP results are all positive. Is that because of contamination?

Ans: Sample designations have been added for Figure 4B of the modified manuscript to
clarify this point as follows; “(i) a BSS coupon remained uninoculated (NC BSS) and were

processed alongside as a negative control; (ii) a swab negative control in DRS (ZS); (iii) a
swab with 5,000 copies of NATtrol in DRS (ZSZ); and (iv) 5,000 copies of NATtrol control
extracted directly from Maxwell (ZPC).” Thus, ZS1 was a negative control, which was
negative for both RT-qPCR and RT-LAMP. ZSZ1 and ZPC are both positives and they are
positive for both RT-qPCR and RT-LAMP.

For BSS2 samples, all replicates were positive for RT-LAMP assay and not for RT-qPCR
assay as pointed out by the reviewer. These products were further sequenced via Sanger
method and were confirmed as the SARS nCoV-2 virus sequence, matching 100% with the
reference sequence. This is not a contamination since all samples, including negative
samples, were handled on the same date by the same personnel. The BSS2 samples were
tested again by RT-LAMP assay and sequence analysis, and were confirmed as SARS nCoV-
2 virus sequence.

12. The authors tried to repeat the detection with LAMP assay, however, it is unclear what
conclusions they were trying to make. It looks like the LAMP has a worse sensitivity than
qPCR, even though in some other reports people claim the LOD is ~2 copies per ul of
samples. I can hardly see how this section is related to the main topic of this paper.

Ans: The use of RT-LAMP is one of several approaches used for routine analysis for
COVID19 and is being used by many investigators for both surveillance and clinical uses. It
provides another legitimate orthogonal tool to ddPCR and RTqPCR for several reasons
including 1) it has higher specificity to the target locus since it requires 6 primers to generate
a amplicon, 2) It targets a longer amplicon fragment (~130 bp) over the CDC method (N1
72-bp, N2 67-bp), 3) its resulting amplicon can be validated using Sanger and other
sequencing methods where the CDC cannot, and 4) it uses a non-PCR approach to detection
independent to unbalanced Tm of primer-probe and annealing challenges that we see with the
CDC method. RT-LAMP followed by Sanger sequencing hybrid approach can be used to
disqualify possible false positives and negatives.

13. Lines 211-219: Did the author test the RNA yield after the extraction? The reason why the
authors saw lower recovery for samples containing environmental debris is probably because
of the low RNA yield for those samples. And the low-yield is not resulted from “inhibition”,
but just because of those debris can physically impact the column’s RNA binding capability
during the extraction. It is a typical step to remove large cell debris or visible solid materials
in the supernatant before transferring lysed samples onto the column during RNA extraction
procedure. However, since the authors were using an automated extraction machine, it didn’t
“notice” this issue.

Ans: The authors did not determine the RNA yield via nanodrop or Qbit assays but we
quantified the RNA copy numbers via RT-qPCR since the RNA content was extremely low,
below the detection limit for either nanodrop or Qbit instrument.

The authors didn’t use an extraction system that used columns as pointed out by the reviewer.
Instead the authors used the Maxwell automated system that uses a magnetic bead extraction,
thus, column inhibition due to environmental debris is not possible.

14. Lines 243: It is unclear why the authors claim the limit is 1000 viral particles per 25 cm².
They only tested 5000 copies in previous sections, and get positive results for some
measurements.

Ans: With the constraints of coupon size and the sample's low viscosity, only 10 aliquots of
10 µL of the viral solution was possible on to the 25 cm² coupon. The maximum number of
viral particles per 10 µL spot was 500 virus particles, otherwise the spots would not get
desiccated over a 24 hours period. In this way we were able to spike ~5,000 viral particles
207 per 25 cm² coupon.

Since under ideal conditions we were able to recover 0.03% and 1.68% of viral particles
(Figure 3), with all the permutations (loss during sampling, transportation, extraction, and
inhibition due to environmental debris) we concluded that the limit of the E2E process in the
viral recovery was 1,000 viral copies per 25 cm².

15. Lines 245-247: why the authors think this combination with DRS medium is the best? In
figure 1, figure 2, and figure 3, the samples resuspended in Water are consistently better than
in DRS medium (100% vs 75%).

Ans: Despite water having a higher yield than DRS in Figures 1 - 3, DRS is capable of
maintaining nucleic acid stability over much longer periods of time at room temperature
during sample transport and storage than if the sample was kept in water. Numerous studies
confirmed that water will not shield RNA molecules at room temperature even for few hours.
Additionally, as mentioned above in question 3, as part of our IRB we had to inactivate any
virus that was collected during the sample collection before transporting to BSL-2 lab. Water
does not inactivate viruses, and it was much more efficient to inactivate the samples in the
DRS transport media rather than adding an inactivation step that would lead to further losses
of sample yield.

16. Lines 248-252: 368 samples were collected from seven different material surfaces in 10
buildings, and found all of them were negative of SARS-CoV-2. Whether they were people
working in the building during the sample collection period? How many people in and out of
the building per day? Whether there was a positive covid19 patients (even related) found in
the buildings? Whether people in the building were required to wear masks? All those
information are necessary in order to interpret the Negative results.

Ans: The reviewer brings up valid questions as to the likelihood that samples may contain
SARS-CoV-2 virus. Environmental sampling was only conducted in buildings that were
actively being utilized and that had personnel. All sampling was performed in these buildings
in the morning prior to the routine daily cleaning of each building to ensure that any SARS-
CoV-2 virions remaining on the surfaces from the previous work day would not be removed
prior to our sampling. Each building varied in the number of active staff members that passed
through on a daily basis. Unfortunately, due to confidentiality concerns and agreements with
facility managers, exact numbers cannot be given; however, we can confer that each building
had anywhere from 1 employee to 10s of employees working across the 10 buildings
sampled in this study. Additionally, due to confidentiality concerns we cannot relate if there
were any COVID19 positive staff members (or family) identified in the buildings during the
time of our sampling. Personal Protective Equipment, including masks and in some case
visors/goggles, were strictly enforced along with frequent hand washing and the sanitization
of surfaces after their use. In addition, "Safe at Work" training was provided to the employee
to follow strict guidelines.

The above information has been added on Line# 558-571.

17. The authors ran technical replicates for qPCR, what if one replicate is positive within the
range of quantification and the other is negative? How was this handled?

**Ans: In the case that one or more of the three qPCR replicates came back positive, but was**
**below the limit of detection (10 viral copies/5 μ L of RNA extract) then the sample would be**
**rerun and the positives would be sent off for sequencing to confirm the samples contain viral**
**material. Figure 4 shows an example of how we handled this scenario. However, none of the**
**368 environmental tested samples tested exhibited such scenarios.**

18. There is no Supplemental Figure 5A or 5B. Does the authors re-check the original positive
controls' concentration before determining the limit of detection?

**Ans: This is an oversight by the authors and is corrected now as Supplemental Figure 5 and 5**
**inset instead of 5A and 5B. About re-checking the original positive controls, in this LoD**
**experiment, synthetic RNA fragment (IDT) was used and not viral standards; hence authors**
**did not re-check the concentration. The authors used fresh IDT synthetic RNA fragments as**
**positive control every single run and avoided any inadvertent degradation by minimizing**
**freeze/thaw. The standard curve with 98 to 99% R^2 value and an efficiency of 90 to 100%**
**were achieved. When there are deviations from these standard curves observed, those**
**datapoints were not used and the experiments were repeated.**

Reviewer #2 (Comments for the Author):

This manuscript describes the systematic examination of the efficiencies of procedures from
sampling to qPCR and the impact of each individual step on the recovery of SARS-CoV-2 from
surfaces. The authors conclude by providing a practical protocol for surface monitoring of SARS-
CoV-2 with its recovery rate and detection limit, which greatly aids the interpretation of results.
The information presented in this study, particularly the granular breakdown of losses at each
step, could also be valuable to the detection of other pathogens from the built environment.
The study is overall thorough, and the results are of immediate interest.

A few specific comments:

It appears that the number of data points collected for different conditions (water vs. DRS,
different surface samples) varied widely. Please comment on the effect of this imbalance on the
statistical analysis. If the variation reflects exclusion of certain datapoints, please explain the
exclusion criteria.

**Ans: The Ct value variability among the technical replicates (n=3) were not large enough so that**
**they would fall within 0.5 Ct value for positive controls (>100 copies/5 μ L RNA extracts).**
**However, the biological replicates showed variability of more than 0.5 Ct values. Here, there is**
**a necessity to convert the Ct values to actual numbers based on the standard curve to calculate**
**percent recovery. Since positive control is high enough in terms of number of copies (>100 per**
**5 μ L), the variation was well within 10% but when sampled from surfaces or in combination of**
**other variable, 20% to 25% deviation from the mean was observed. When the values were**
**more than 25%, we calculated outliers and removed from plotting in the graphs and also from**
**the calculations. In general 3 technical replicates and 3 biological replicates were used which**
**will give 9 datapoints. Outliers were screened using the rOut method from the robustX R**
**package (<https://CRAN.R-project.org/package=robustX>). The statistical analysis used for outlier**
**screening (interquartile range method) was given in the manuscript (Line 630 to 631).**

For longer term impact, considering providing some details on the proprietary swab types and
kits used (e.g., materials or important properties of the kit), such that the overall findings might
still be useful after the manufacturers change their design or become defunct.

**Ans: The authors agree with the reviewer that presenting identifying information about the**
**protocols and tools/consumables is critical to reproducible data, and especially important in a**
**research article that is designing and evaluating a protocol. The authors took extreme care and**
**contacted vendors to provide necessary details of the products. Some information was not**
**available despite our requests but most of them were given that were made available to us.**

I. 69 Specify which government.

**Ans: The authors have specified the government of the United States.**

II. 86-88 A monitoring "plan" might include details like which surfaces should be sampled and
how frequently. More accurately, this study outlines a comprehensive method that can be used
in environmental monitoring. Please rephrase.

**Ans: The authors have rephrased the sentence to read, "In this study we outline a**
**comprehensive approach to characterize and develop an effective environmental monitoring**

methodology that can be used to better understand viral persistence in built environments, and
aid in the virus' elimination."

I. 115 Over how much time? Are results likely to vary over time? Same for temperature.

Ans: The manufacture claims that their proprietary transport medium (DRS) can keep RNA
stable for ≥ 1 month when kept between 25°C and 4°C, and keep RNA stable for ≥ 2 years when
kept between 4°C and -25°C. It is likely that over extended periods of time and at elevated
temperatures RNA would become degraded. However, in this study once samples were
collected, they were kept insulated from higher than room temperature heat, and subsequently
processed within 3 hours of collection.

Since we have not tested these in our laboratory during this study, we are not incorporating
these results in the manuscript.

I. 319 Intriguing hypothesis. Why not measure roughness and hydrophobicity here?

Ans: "The authors agree that an in-depth analyses of the materials' surfaces microstructures
would likely shed light on this phenomenon observed with PETG. Unfortunately, this degree of
surface testing is outside the scope of this paper."

I. 333 This is an important point that bears emphasizing. Please comment on the potential
implications of RNA-based detection methods for assessing disinfection, which largely targets
membrane/capsid integrity. Perhaps also comment on the potential for additional assessment
of integrity (e.g., <https://www.medrxiv.org/content/10.1101/2020.05.08.20095364v1>).

Ans: As mentioned, it is possible and presumed that the disinfection process degenerated RNA
molecules but still left behind the smaller residual fragments. The 80-bp target of the RT-qPCR
assay employed during this study might render the virus non-viable due to the disinfectant
bleach. Use of PMA as mentioned in the medrxiv article that reviewer suggested might be
useful. Since 2005 our group has used PMA technology to remove naked nucleic acids and
compromised microbial cells from PCR amplification. We also tested herpes virus, a DNA virus,
successfully using PMA method and measured viable viral particles using RT-qPCR
(unpublished). We included the following reference where use of PMA technique in measuring
aquatic viral population was peer-reviewed and published.

Randazzo, W., Khezri, M., Ollivier, J., Le Guyader, F.S., Rodriguez-Diaz, J., Aznar, R., Sanchez, G.,
2018a. Optimization of PMAxx pretreatment to distinguish between human norovirus with
intact and altered capsids in shellfish and sewage samples. Int J Food Microbiol 266, 1-7.

*"In order to avoid these false positive results after the bleach treatments, samples*
*should be tested using an alternative technique that targets longer RNA fragments, such*
*as RT-LAMP. In addition nucleic acids intercalating dyes were reported to be useful in*
*eliminating naked nucleic acids and compromised microbial structures for bacteria [1],*
*fungi [2], and DNA/RNA viruses [3]."* See Lines 337 to 341.

II. 325-326 What is meant by the chemical nature of the paint? Please elaborate.

Ans: In this study we used bare 302 stainless steel (BSS), painted 302 stainless steel (PSS; white
acrylic paint 168130-Rust-Oleum, Vernon Hills, IL), polyethylene terephthalate modified with
glycol (PETG), and fiberglass-reinforced plastic (FRP) (See Lines 500 to 503).

The chemical characteristics are different from each other. For example, PETG contain same
monomers as that of PET, but also includes glycol, giving it different chemical properties than
PET. Similarly the painted surfaces have acrylic compound (includes acrylic acid or related
compounds called acrylates) that may interact differently with the bleach and the viral particles
than bleach and the other surface materials.

Having all these chemical properties written in the manuscript is possible, if Editor is interested
to add them.

II. 545-547 Rephrase: "Field control samples were collected in a similar manner to
environmental samples, but instead..."

Ans: The authors have rephrased the sentence to read (Line 553 to 556),

"Environmental sampling of built environment surfaces was conducted in an identical
manner, except that the moistened swab for the field control was not touched to a
surface, but rather was waved in the air for 2 min prior to breaking off the swab head
into a barcoded DRS tube."

I. 551 What type of debris? Where did these samples come from?

Ans: The environmental debris mentioned in this section, and throughout the paper, refers to
dust and other small particulate matter that accumulate on surfaces. Sampling locations were
addressed on lines 251 to 252, "The samples collected were from seven different materials
found in 10 buildings, including stainless steel, Amerstat, plastic, copper, and painted surfaces."

The authors have added the additional clarification, "All selected surfaces were in areas of the
facilities with large amounts of pedestrian traffic, and deemed as high touch surfaces capable of
serving as SARS-CoV-2 fomites." Line 253 to 254.

I. 571 This section needs more detail. Again, where did samples come from? Define what
controls were included.

Ans: Please see the above addition clarifying the surfaces sampled line 251 to 252.

Environmental controls details are given in Lines 553 to 556.

Similarly, some more details were given about environmental sample collection in Lines 558 to
571. Specific name of the location and building are prohibited. But these were from a typic
work place environment spanning from work table, floor, corridor, handle, lift operation
buttons, rest rooms, dining area, etc. These details can be added, if needed.

Typos and other errors: I.73, I. 78, I. 282

Ans: The authors have corrected the errors and typos. Missing reference in line 282 is added
[4].

- 1. Vaishampayan P, Probst AJ, La Duc MT, Bargoma E, Benardini JN, Andersen GL,
Venkateswaran K: **New perspectives on viable microbial communities in low-biomass**
**cleanroom environments.** *ISME J* 2013, **7**(2):312-324.
- 2. Vesper S, McKinstry C, Hartmann C, Neace M, Yoder S, Vesper A: **Quantifying fungal**
**viability in air and water samples using quantitative PCR after treatment with**
**propidium monoazide (PMA).** *J Microbiol Meth* 2008, **72**(2):180-184.
- 3. Randazzo W, Khezri M, Ollivier J, Le Guyader FS, Rodríguez-Díaz J, Aznar R, Sánchez G:
**Optimization of PMAxx pretreatment to distinguish between human norovirus with**
**intact and altered capsids in shellfish and sewage samples.** *Int J Food Microbiol* 2018,
**266**:1-7.
- 4. Nagura-Ikeda M, Imai K, Tabata S, Miyoshi K, Murahara N, Mizuno T, Horiuchi M, Kato K,
Imoto Y, Iwata M *et al*: **Clinical Evaluation of Self-Collected Saliva by Quantitative**
**Reverse Transcription-PCR (RT-qPCR), Direct RT-qPCR, Reverse Transcription–Loop-**
**Mediated Isothermal Amplification, and a Rapid Antigen Test To Diagnose COVID-19.** *J*
*Clin Microbiol* 2020, **58**(9):e01438-01420.

September 22, 2020

Dr. Kasthuri Venkateswaran
California Institute of Technology
Jet Propulsion Laboratory
Mail Stop 245-105
4800, Oak Grove Dr.
Pasadena, CA 91109

Re: mSystems00771-20R1 (End-to-End Protocol for the Detection of SARS-CoV-2 from Built Environments)

Dear Dr. Kasthuri Venkateswaran:

Your manuscript has been accepted, and I am forwarding it to the ASM Journals Department for publication. For your reference, ASM Journals' address is given below. Before it can be scheduled for publication, your manuscript will be checked by the mSystems senior production editor, Ellie Ghatineh, to make sure that all elements meet the technical requirements for publication. She will contact you if anything needs to be revised before copyediting and production can begin. Otherwise, you will be notified when your proofs are ready to be viewed.

Sincerely,

Sean Gibbons
Editor, mSystems

Journals Department
Supplemental Figure 1: Accept

Supplemental Figure 3: Accept

Data Set-1: Accept

Supplemental Material Table 2: Accept

Supplemental Figure 6: Accept

Supplemental Figure 5: Accept

Supplemental Figure 2: Accept

Supplemental Figure 4: Accept

Supplemental Table 1: Accept